# Physics-Informed Residual Flows

**Jephte Abijuru** [1]  **Mayank Nagda** [1]  **Phil Sidney Ostheimer** [1]
**Sebastian Josef Vollmer** [1,2]  **Marius Kloft** [1]  **Sophie Fellenz** [1]

## Abstract

Physics-Informed Neural Networks (PINNs) embed physical laws into deep learning models. However, conventional PINNs often suffer from failure modes leading to inaccurate solutions. We trace these failure modes to two structural pathologies: gradient shattering, where gradients degrade with depth and provide little training signal, and flow mismatch, where training pushes predictions along trajectories that diverge from the PDE solution path. We introduce ResPINNs, which reformulate PINNs as residual flows, networks that iteratively refine their own predictions through explicit corrective steps, in the spirit of classical iterative solvers. Our analysis shows that this design mitigates both pathologies by keeping updates aligned with descent and by preserving informative gradients across depth. Extensive experiments on PDE benchmarks confirm that ResPINNs achieve higher accuracy with substantially fewer parameters than conventional architectures.

## 1. Revisiting Failure Modes in PINNs

Partial differential equations (PDEs) govern a wide range of physical, engineering, and scientific systems. Because closed-form solutions are rarely available, numerical solvers such as finite difference, finite element, or spectral methods are the standard tools, but these approaches are computationally costly and restricted to discretized meshes. Physics-Informed Neural Networks (PINNs) (Raissi et al., 2019) have emerged as a promising alternative, embedding PDE, initial, and boundary conditions into the neural network loss function. By leveraging automatic differentiation, PINNs can approximate PDE solutions continuously in space and time.

Despite recent advances, PINNs remain vulnerable to intrinsic failure modes. Krishnapriyan et al. (2021) documents several types of PDEs that are especially challenging, often due to parameters that induce high-frequency or complex solution behaviors. In such cases, PINNs may fail to accurately propagate initial conditions. A typical manifestation is the emergence of overly smooth solutions that minimize empirical loss while ignoring temporal dynamics.

To address these shortcomings, various strategies have been proposed, including optimization techniques (Wu et al., 2024; Wang et al., 2021; Liu et al., 2025; Wong et al., 2022; Bu & Karpatne, 2021), adaptive sampling (Daw et al., 2023), architectural modifications such as sequence-based models (Zhao et al., 2024; Xu et al., 2025a; Wang et al., 2021) and approaches emphasizing residual corrections like PirateNets (Wang et al., 2024). However, these methods do not explicitly address the instability caused by noisy gradients, often referred to as gradient shattering, which can mislead training and limit robustness.

Orthogonal to these advances, we revisit the failure modes of PINNs from two complementary angles: optimization dynamics and representation flow. We identify two structural problems: (i) *gradient shattering*, where increasing depth causes input–output Jacobians to decorrelate exponentially and their norms to vanish or explode; and (ii) *flow mismatch*, where training updates in latent space do not align with true descent directions, allowing the network to satisfy residual constraints locally while drifting globally and failing to propagate initial conditions. Since PDE residuals require repeated differentiation of network outputs, gradient shattering is amplified in PINNs, making optimization unstable even when the residual loss is small.

To address these issues, we introduce *residual flows*: networks designed as iterative refinement schemes, where each component performs a small correction around the identity. This stepwise view connects directly to three established perspectives: (i) residual networks, where skip connections stabilize gradients; (ii) neural ODEs, where depth corresponds to integrating a continuous-time flow; and (iii) classical iterative solvers, where predictor–corrector updates progressively reduce error. In the PINN setting, these formulations coincide: residual flows stabilize Jacobians, keep

---

[1]Department of Computer Science, RPTU University Kaiserslautern-Landau [2]DSA, German Research Center for AI. Correspondence to: Jephte Abijuru <abijuru@rptu.de>.

*Proceedings of the 43rd International Conference on Machine Learning*, Seoul, South Korea. PMLR 306, 2026. Copyright 2026 by the author(s).

updates aligned with loss descent, and preserve initial and boundary conditions across depth.

Our paper makes the following contributions:

1. We analyze both empirically and theoretically why standard PINNs fail, tracing condition-propagation errors to *gradient shattering* and *flow mismatch*.

2. We propose *Residual Flows*, which view solution learning of PDEs as stepwise refinement via small residual corrections around the identity, aligning PINNs with classical predictor–corrector methods.

3. Through theory and ablations, we show that residual pathways, rather than explicit sequence modules, are the stabilizing mechanism behind recent performance gains. Empirical results on convection, reaction, and wave PDEs confirm improved condition preservation and solution fidelity.

## 2. Dissecting Existing Approaches to Failure Modes

A central challenge in PINNs for time-dependent PDEs is the propagation of initial conditions across time. Several recent works have sought to address this by introducing explicit sequence modeling. For example, Krishnapriyan et al. (2021) proposed recursive sequence-to-sequence training, rolling solutions forward in time with separate networks. While effective for short horizons, this strategy is memory- and compute-intensive, and does not generalize reliably outside the training window. More recent approaches have adapted modern sequence architectures: Zhao et al. (2024) introduced a transformer-based framework (PINNsFormer), while Xu et al. (2025a) proposed state-space models (PINN-Mamba). Both report improved accuracy and robustness, attributing their gains to the ability of attention or structured recurrence to capture long-range temporal dependencies.

At first glance, these results seem to suggest that sophisticated sequence modules are essential for overcoming failure modes in time-dependent PINNs. Yet this conclusion is not entirely satisfying: improvements could equally stem from side effects such as increased parameterization, altered optimization dynamics, or more flexible local mappings. In other words, what appears as a benefit of "long-range temporal modeling" may instead be an artifact of broader architectural changes. This motivates a sharper question: *are sequence modules truly the driving factor behind the reported improvements, or are we attributing gains to the wrong mechanism?* To probe this, we designed a controlled ablation study. Training setup, initialization, and sampling were kept fixed, and only the internal sequence modules were varied. Specifically, self-attention and state-space operators were replaced with deliberately simple local map-

*Table 1.* Ablation study on Convection, Reaction and Wave PDEs. The relative MAE (rMAE) values are reported. The "–" symbol indicates a removed component and "+" indicates a replacement. For example, *PINNsFormer –Attention+MLP* replaces the attention block with an MLP. Removing attention and replacing it with linear mappings preserves or even improves performance, despite the drastic reduction in complexity. Lower is better.

| Model | Convection | Reaction | Wave |
|---|---|---|---|
| PINNsFormer (Original) | 0.510 | 0.015 | 0.270 |
| -Decoder | 0.043 | 0.017 | 0.058 |
| -Attention + Linear | 0.012 | 0.022 | 0.022 |
| -Attention + MLP | 0.009 | 0.016 | 0.142 |
| PINNMamba (Original) | 0.019 | 0.010 | 0.020 |
| -SSM | 0.012 | 0.013 | 0.029 |
| -SSM+MLP | 0.063 | 0.014 | 0.015 |

pings (a linear projection or a shallow MLP), with parameter counts carefully matched within $\pm 10\%$. This isolates the effect of explicit sequence modeling from confounding factors such as model capacity or optimization differences.

The evidence in Table 1 challenges the conventional explanation. The table compares both transformer-based (PINNs-Former) and state-space–based (PINNMamba) architectures against ablated versions where their sequence modules are removed or replaced with simpler alternatives. For PINNs-Former, the encoder is retained while attention is stripped out and substituted either with a linear projection or a shallow MLP. For PINNMamba, the state-space operator is removed outright or replaced by an MLP with matched parameter count. Across all three PDE benchmarks, these simplified variants perform comparably to the original models, indicating that explicit attention or structured recurrence is not essential for maintaining accuracy. This suggests that the improvements attributed to sophisticated sequence modules may instead arise from a different architectural mechanism.

Across both transformer- and state-space–based PINNs, one component remains consistent: the use of *residual pathways* that carry predictions forward through incremental corrections. Unlike attention or structured recurrence, these pathways are present in every variant tested, including the simplified ablations. This observation points to residual connections—not sequence modules—as the common mechanism underlying stability and accuracy.

Why might residual pathways play such a critical role? At a high level, they enforce an update rule that keeps each layer close to the identity, nudging predictions forward through small, controlled steps rather than drastic transformations. This structure has several consequences that help explain the observed robustness:

(H1) Because updates are incremental, optimization becomes more stable: each layer only needs to make small corrections, reducing the risk of divergence.

(H2) The skip connections implicit in residual design bias the layer Jacobians toward the identity, which mitigates gradient shattering and helps preserve information across depth.

(H3) The repeated corrections accumulate like iterations of a solver, progressively refining the solution in the manner of predictor–corrector schemes.

Taken together, these hypotheses recast the source of robustness in time-dependent PINNs: not the sophistication of sequence modules, but the refinement dynamics induced by residual flows. In the remainder of this paper, we put these hypotheses to the test.

# 3. Mitigating Failure Modes with Residual Alignment

PINNs can achieve low training loss yet still produce drifting solutions. We trace this to two mechanisms: *gradient shattering*, where Jacobians lose coherence and their norms vanish or explode with depth, degrading the derivative signal that PINNs rely on; and *flow mismatch*, where training pushes predictions along trajectories that diverge from the PDE solution. To address these issues, we view training not as a single mapping but as an *evolving flow in latent space*, advanced step by step through small residual updates. This perspective makes explicit two stabilizing principles: (i) alignment of updates with descent directions, and (ii) near-identity Jacobians that preserve gradient propagation. We begin by analyzing gradient shattering and then show how residual formulations encourage alignment.

## 3.1. Preliminaries

We consider PDEs on a spatio–temporal domain $\Omega \times [0, T]$ with solution $u : \Omega \times [0, T] \to \mathbb{R}^m$ subject to interior, initial, and boundary operators $\mathcal{F}, \mathcal{I}, \mathcal{B}$:

$$\mathcal{F}(u)(x,t) = 0, \quad \mathcal{I}(u)(x,0) = 0, \quad \mathcal{B}(u)(x,t) = 0,$$

PINNs (Raissi et al., 2019) approximate $u$ by a neural network $u_\theta$ and train by minimizing residuals at collocation points: interior $\mathcal{X} \subset \Omega \times (0, T]$, initial $\mathcal{X}_0 \subset \Omega \times \{0\}$, and boundary $\mathcal{X}_\partial \subset \partial\Omega \times [0, T]$. The objective is a weighted mean–squared residual,

$$
\begin{aligned}
L(u_\theta) = &\tfrac{\lambda_\mathcal{F}}{|\mathcal{X}|} \sum_{(x,t)\in\mathcal{X}} \|\mathcal{F}(u_\theta)(x,t)\|^2 \\
&+ \tfrac{\lambda_\mathcal{I}}{|\mathcal{X}_0|} \sum_{(x,0)\in\mathcal{X}_0} \|\mathcal{I}(u_\theta)(x,0)\|^2 \\
&+ \tfrac{\lambda_\mathcal{B}}{|\mathcal{X}_\partial|} \sum_{(x,t)\in\mathcal{X}_\partial} \|\mathcal{B}(u_\theta)(x,t)\|^2,
\end{aligned}
\tag{1}
$$

where $\lambda_\mathcal{F}, \lambda_\mathcal{I}, \lambda_\mathcal{B} \geq 0$ balance the constraints.[1]

## 3.2. Gradient Misalignment in PINNs

A well-documented pathology in deep networks is *gradient shattering*: correlations between input–output sensitivities at nearby inputs decay exponentially with depth, while their norms either vanish or explode depending on initialization scaling (Balduzzi et al., 2017; Poole et al., 2016; Pennington et al., 2018; Yang & Schoenholz, 2017). Since PINNs embed PDE residuals into the training objective only at sparse collocation points, they are especially vulnerable to this effect: low residuals can coexist with large solution drift between points. To formalize this, let $J_\theta(x,t) = \nabla_{(x,t)} u_\theta(x,t) \in \mathbb{R}^{m\times(d+1)}$ denote the input–output Jacobian of the network. We summarize the mean-field behavior below.

**Theorem 3.1** (Informal; mean-field gradient shattering). *Let $u_\theta$ be a depth-$L$, width-$n$ fully connected PINN with i.i.d. Gaussian initialization, a 1-Lipschitz activation and network parameters $\theta$. Denote its Jacobian $J_\theta(z) = \nabla_z u_\theta(z)$ at input $z = (x,t)$. For nearby $z'$ with $\|z' - z\| \leq r_0$, define the Frobenius cosine $\cos(J_1, J_2) = \langle J_1, J_2 \rangle_F / (\|J_1\|_F \|J_2\|_F)$. In the mean-field limit $n \to \infty$:*

*(A) (Exponential decorrelation) $\mathbb{E}[\cos(J_\theta(z), J_\theta(z'))] = \mathcal{O}(\rho^L)$ for some $\rho \in (0, 1)$.*

*(B) (Norm growth/decay) $\mathbb{E}\|J_\theta(z)\|_F^2 = \Theta(\gamma^L)$ for some $\gamma > 0$, with $\gamma = 1$ only at critical variance.*

*Thus, unless tuned to the edge of chaos, Jacobians decorrelate exponentially and their norms vanish or explode with depth.*

A detailed statement and proof, adapted from classical mean-field analyses of deep random networks, is provided in Appendix D.

**Implications for PINNs.** Sparse collocation makes PINNs particularly vulnerable to gradient shattering: while residuals may vanish at training points, exponential loss of

---

[1] We use $\langle A, B \rangle_F = \mathrm{tr}(A^\top B)$, $\|A\|_F$ for Frobenius norms, $\|\cdot\|$ for Euclidean norms, and $\|\cdot\|_2$ for spectral norm.

Jacobian correlation and unstable norms (Theorem 3.1) allow the learned solution to drift in between. This motivates enforcing near-identity Jacobians and residual alignment mechanisms to stabilize training. Yet gradient shattering alone only explains how depth degrades the derivative signal; it does not address how individual network updates contribute to optimization. To examine this, we turn to *flow mismatch*, focusing on whether layerwise transformations align with descent directions.

### 3.3. Flow Mismatch can hurt PINNs

To understand the gradient misalignment associated with PINNs, we intrepret training as a *latent-space flow problem* indexed by an auxiliary *solver time* $k$:

$$\frac{dz(k)}{dk} = T\big(z(k), k; x, t\big), \quad k \in [0, K], \quad z(0) = E(x, t),$$
(2)

where $z(k) \in \mathbb{R}^{d_h}$ is a latent state obtained from the encoding $E(x, t)$, and $T : \mathbb{R}^{d_h} \to \mathbb{R}^{d_h}$ denotes the residual transformation that advances $z(k)$ toward the PDE solution. This operator may be fixed (as in classical solvers) or learned (as in neural architectures introduced later). In discrete form,

$$z_{k+1} = z_k + T_k\big(z_k; \alpha\big), \qquad k = 0, \ldots, K - 1,$$

with step parameter $\alpha > 0$ implicit in $T_k$. When $\|T_k\|$ is small, each update is a residual correction around the identity. This lens makes two optimization effects explicit: (i) *iterative refinement*, where many small, well-aligned corrections reduce the loss predictably; and (ii) *Jacobian neutrality*, where near-identity Jacobians stabilize gradient propagation across depth. Consider a composition of $K$ such transformations and a PINN loss function $\mathcal{L}(z_k)$ on the $k^{th}$ transformation following the definition in Eq. 1 (Section 3.1).

**Lemma 3.2** (Local update descent with depth-aware smoothness). *Let $z_{k+1} = z_k + T_k(z_k)$ with Jacobian $J_k := \partial z_{k+1}/\partial z_k = I + A_k$. If $\mathcal{L}$ has $\beta$-Lipschitz continous gradient in a neighborhood of $z_k$, then there exist*

$$\beta_k \leq \beta \Big( \prod_{\ell=k}^{K-1} \|J_\ell\|_2 \Big)^2$$

*such that $\mathcal{L}(z_{k+1}) \leq \mathcal{L}(z_k) + \langle \nabla_{z_k}\mathcal{L}(z_k)^\top, T_k \rangle + \frac{\beta_k}{2}\|T_k\|^2$.*

This follows from a first-order Taylor expansion of the $\mathcal{L}(z_{k+1}) := \mathcal{L}(z_k + T(z_k))$. The formal statement and proofs are deferred to the Appendix C. Notice that a first order term is a good approximation when the magnitude of $T_k$ is small enough. Rolling out Lemma 3.2 over

$k = 0, \ldots, K-1$ updates recursively gives

$$\mathcal{L}(z_K) \leq \mathcal{L}(z_0) + \sum_{k=0}^{K-1}\Big[\langle \nabla_{z_k}\mathcal{L}(z_k)^\top, T_k \rangle + \frac{\beta_k}{2}\|T_k\|^2\Big].$$

In particular, Lemma 3.2 implies that the first-order change in the loss at step $k$ is driven by the dot product between the local loss gradient and $T_k$,

$$\mathcal{L}(z_{k+1}) - \mathcal{L}(z_k) \approx \langle \nabla_{z_k}\mathcal{L}(z_K), T_k(z_k)\rangle.$$

We can characterize this via *gradient alignment*: the cosine between the step and the negative gradient,

$$\mathrm{GA}_k := \frac{\langle T_k(z_k), -\nabla_{z_k}\mathcal{L}(z_k)\rangle}{\|T_k(z_k)\|\,\|\nabla_{z_k}\mathcal{L}(z_k)\|},$$

so that

$$\langle \nabla_{z_k}\mathcal{L}(z_k), T_k(z_k)\rangle = -\|T_k(z_k)\|\,\|\nabla_{z_k}\mathcal{L}(z_k)\|\,\mathrm{GA}_k.$$

Thus, each local update constitutes a gradient-based step whose contribution is exactly proportional to its alignment with $-\nabla_{z_k}\mathcal{L}(z_k)$: $\mathrm{GA}_k > 0$ moves $z_k$ into the descent half-space (first-order decrease), $\mathrm{GA}_k = 0$ is neutral, and $\mathrm{GA}_k < 0$ moves uphill.

For PINNs, however, alignment with the aggregate loss gradient is not sufficient to guarantee that all physical constraints are enforced. At the parameter level, the PINN objective is a soft-constrained multi-objective loss

$$\mathcal{L}(\theta) = \sum_{m=1}^{M} \lambda_m \mathcal{L}_m(\theta),$$
(3)

where each component $\mathcal{L}_m$ corresponds to a PDE residual, boundary condition, initial condition, data term, or auxiliary constraint. Let

$$g_m(\theta) := \nabla_\theta \mathcal{L}_m(\theta), \qquad g(\theta) := \nabla_\theta \mathcal{L}(\theta) = \sum_{m=1}^{M} \lambda_m g_m(\theta).$$
(4)

Although the optimizer follows the aggregate descent direction $-g(\theta)$, each constraint induces its own descent direction $-g_m(\theta)$. We therefore define the constraint-level alignment as

$$a_m(\theta) := \frac{\langle g(\theta), g_m(\theta)\rangle}{\|g(\theta)\|\,\|g_m(\theta)\|}.$$
(5)

For a sufficiently small gradient step $\theta^+ = \theta - \eta g(\theta)$, the first-order change in the $m$-th constraint loss satisfies

$$\mathcal{L}_m(\theta^+) - \mathcal{L}_m(\theta) = -\eta\langle g_m(\theta), g(\theta)\rangle + O(\eta^2).$$
(6)

Hence, if $a_m(\theta) < 0$, the aggregate descent direction increases the $m$-th constraint loss to first order, even if the total

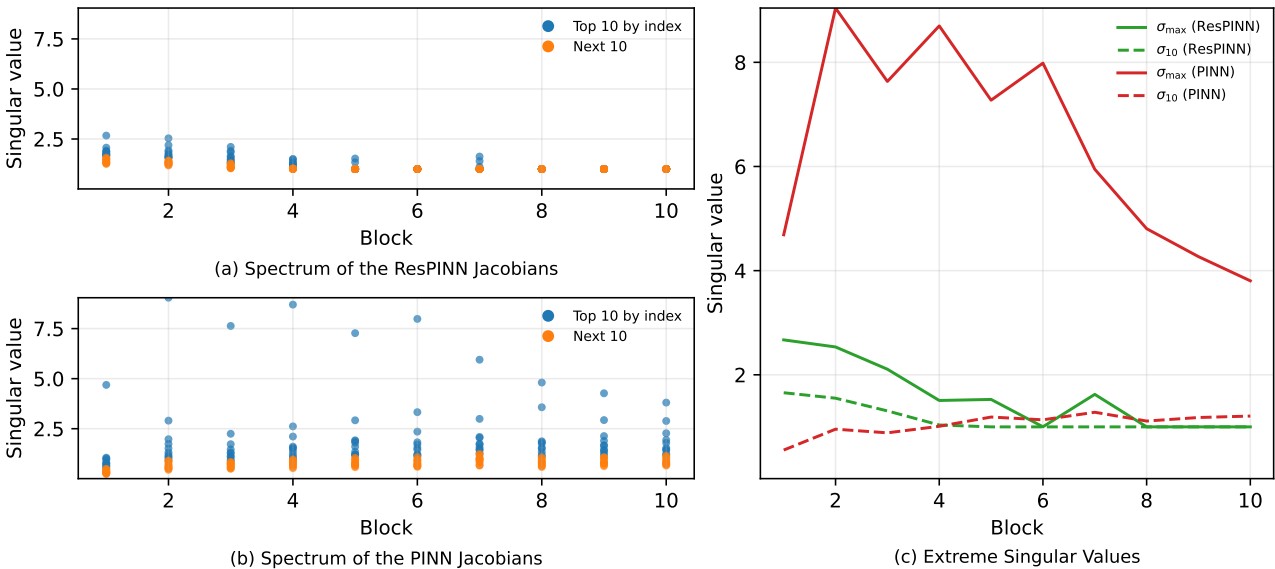

*Figure 1.* Spectral analysis of Jacobians across network depth for the 1D convection problem. (a) ResPINN (with residual connections): singular values remain clustered near unity, indicating near-isometric mappings and stable gradients. (b) Standard PINN: singular values remain spread with depth, reflecting anisotropy and poor conditioning. (c) Extreme singular values ($\sigma_{\max}$ and $\sigma_{10}$) highlight the contrast: PINNs amplify dominant directions, whereas ResPINNs suppress spectral growth. Both architectures use the same number of parameters; the only difference is the residual connection inserted after every two layers.

weighted loss decreases. We refer to this regime as *flow mismatch*: the aggregate descent flow decreases the scalar PINN objective while one or more constraint-level alignments are negative. In PINNs, this is problematic because the individual objectives encode necessary physical constraints, so decreasing the scalar objective does not necessarily imply that the learned trajectory follows a solution-consistent path.

However, alignment alone is not sufficient: even well-aligned updates can suffer from vanishing or exploding gradients if Jacobian spectra are uncontrolled. As a complementary effect, residual formulations also encourage near-identity Jacobians, as shown next.

**Theorem 3.3** (Local Jacobian Neutrality). *If $J_k = I + A_k$ with $\|A_k\|_2 \leq \alpha_k < 1$, then*

$$1 - \alpha_k \leq \sigma_{\min}(J_k) \leq \sigma_{\max}(J_k) \leq 1 + \alpha_k,$$
$$\text{and} \quad \kappa(J_k) \leq \frac{1 + \alpha_k}{1 - \alpha_k},$$

*and for all $v$, $(1 - \alpha_k)\|v\| \leq \|J_k^\top v\| \leq (1 + \alpha_k)\|v\|$. $\alpha_k$ denotes an upper bound on the spectral norm of the deviation from identity at step $k$.*

*Proof sketch.* Weyl's inequality gives $|\sigma_i(J_k) - 1| \leq \|A_k\|_2$; the bounds follow immediately.

Thus, *if* residual updates are small (small $\alpha_k$), per-step Jacobians remain near identity, stabilizing gradient propagation across depth. Residual flow formulations exhibit the

three predicted properties: they maintain positive gradient alignment, have near-identity Jacobians, and operate in the small-step regime. Figure 1 (and Appendix 7) illustrates this empirically. This aligns with earlier observations that residual connections implement iterative inference (Greff et al., 2017; Jastrzebski et al., 2018).

*Appendix roadmap.* Appendix C: proofs of Lemma 3.2 and Theorem 3.3, plus guarantees for residual flows. Appendix D: note on gradient shattering.

## 4. Related Work

**Neural Operators.** Neural operators such as DeepONet (Lu et al., 2021a) and FNOs (Li et al., 2023) are data driven *surrogate models* that approximate the PDE solution operator from labeled data, whereas PINNs rely on enforcing PDE residuals and boundary/initial conditions. Since our work focuses on PINNs, we benchmark mainly against PINN variants and restrict our analysis to failure modes specific to this class of methods.

**Flows in Machine Learning.** The idea of flows is well-established in Machine Learning. Normalizing flows learn invertible maps that transform a base density into a data density by composing simple bijections and tracking Jacobian determinants for exact likelihoods (Rezende & Mohamed, 2015; Dinh et al., 2017; Kingma & Dhariwal, 2018; Chen et al., 2018). This probabilistic goal is orthogonal to ours:

we do not model densities or require invertibility. In our work, the term flow is used as a unifying tool for both residuals connections and neural ODEs.

**Residual networks and connections.** Residual connections stabilize training by composing near-identity transformations (He et al., 2016; Greff et al., 2017; Jastrzebski et al., 2018; Haber & Ruthotto, 2017; Lu et al., 2018; Chen et al., 2018). In PINNs, they appear in several proposals, often alongside other architectural changes, so their specific contribution is unclear (Wang et al., 2020; Zhao et al., 2024; Xu et al., 2025a). PINNsFormer (Zhao et al., 2024) and PINN-Mamba (Xu et al., 2025a) both employ residual connections but attribute gains to sequence modeling, while PirateNets (Wang et al., 2024) explore adaptive residual scaling and physics-informed initialization without connecting them to failure modes.

**Continuous-depth limits and Neural ODEs.** Taking residual networks to the continuous-depth limit yields Neural ODEs, parameterized by vector fields and solved numerically (Chen et al., 2018). Connections to solver stability and residual architectures have been emphasized (Lu et al., 2018; Haber et al., 2019), and continuous-depth models have been adapted to scientific machine learning (Yin et al., 2023; Verma et al., 2024).

**Positioning.** Our contribution differs from likelihood-based flows and prior PINN adaptations. We explicitly characterize two structural failure modes in PINNs, gradient shattering and flow mismatch, and propose a residual flow formulation that enforces gradient alignment and Jacobian neutrality. The mechanism is architecture-agnostic: it can be instantiated as a residual stack, as a continuous-depth Neural ODE with explicit solvers, or as a purely iterative refinement scheme without ODE machinery. This unifies discrete residual nets, continuous flows, and solver-style iterations under a single stabilization principle tailored to PINNs.

## 5. Empirical Evaluation

**Architectures.** Our proposal is to view PINNs through the lens of *residual flows*: neural networks that refine predictions iteratively, analogous to numerical solvers advancing a state over time. To investigate this perspective, we consider three architectural variants that will serve as the basis of our analysis (described below):

- **Residual Networks (ResPINN).** Discrete residual flows, where each block applies a correction around the identity $h_{k+1} = h_k + f(h_k; \theta_k)$ with $h_k$ denoting the hidden state (block input/output) at depth $k$, and $f(\cdot; \theta_k)$ the learned transformation within that block.

- **Neural ODEs (O-PINN).** Continuous residual flows obtained in the infinitesimal-step limit, integrating $\dot{h} = f_\theta(h, t)$ with a numerical ODE solver (Chen et al., 2018). This provides the continuous-depth analogue of the residual formulation.

- **Progressive Residual Flows.** A curriculum-style variant of residual networks that increases depth gradually during training by appending new residual blocks while freezing earlier ones. This mirrors multistage solvers where successive corrections extend accuracy.

**Benchmarks.** We evaluate on four established benchmarks. First, three canonical time-dependent PDEs—Wave, Reaction, and Convection—are widely used to probe optimization behavior in PINNs (Raissi et al., 2019; Krishnapriyan et al., 2021; Zhao et al., 2024; Wu et al., 2024) and Heat Equation. Prior work has shown that Reaction–Diffusion and Convection in particular expose common failure modes of PINNs (Krishnapriyan et al., 2021). Moreover, we include the *PINNacle* suite (Zhongkai et al., 2024), a collection of 16 diverse PDE tasks spanning Burgers, Poisson, Heat, Navier–Stokes, Wave. Detailed formulations, discretizations, and training domains are given in Appendix E.

**Baselines.** We compare against a broad suite of PINN architectures, spanning classical approaches (MLP-based PINNs (Raissi et al., 2019), FLS (Wong et al., 2022), QRes (Bu & Karpatne, 2021)), recent improvements (KANs (Liu et al., 2025), RoPINNs (Wu et al., 2024)), state-of-the-art sequential models (PINNsFormer (Zhao et al., 2024), PINNMamba (Xu et al., 2025a)) and two methods closely related to our residual-flow perspective: PirateNet (Wang et al., 2024), which employs adaptive residual networks, and PINN_SOAP (Wang et al., 2025), which stabilizes training via gradient-alignment–based second-order optimization. This collection includes both pointwise networks and methods explicitly designed to address failure modes in dynamical systems using sequence modeling approaches. While PINNsFormer, PINNMamba, and PirateNet all employ residual connections, their motivations differ considerably. PirateNet is the most closely related to our approach; it also employs gating, physics-informed initialization, adaptive loss weighting, weight factorizations, and input feature processing.

**Implementation.** We instantiate the latent *residual flow* architectures as block-structured networks. Unless otherwise noted, all models are trained on $101 \times 101$ space–time grids using the L–BFGS optimizer and the wavelet activation of Zhao et al. (2024). For the baselines, we follow the original configurations: *PINNMamba* uses subsequences of length 7 with step size $10^{-2}$, and *PINNsformer* uses subsequences of length 5 with step size $10^{-4}$. All other models

*Table 2.* Quantitative results on four PDE benchmarks. ResPINN consistently outperforms baselines.

| Model | Wave | | Reaction | | Convection | | Heat | |
|---|---|---|---|---|---|---|---|---|
| | rMAE | rRMSE | rMAE | rRMSE | rMAE | rRMSE | rMAE | rRMSE |
| PINNs | 0.4101 | 0.4141 | 0.9803 | 0.9785 | 0.8514 | 0.8989 | 0.8956 | 0.9404 |
| QRes | 0.5349 | 0.5265 | 0.9826 | 0.9830 | 0.9035 | 0.9245 | 0.8381 | 0.8800 |
| FLS | 0.1020 | 0.1190 | 0.0220 | 0.0390 | 0.1730 | 0.1970 | 0.7491 | 0.7866 |
| PINNsFormer | 0.3559 | 0.3632 | 0.0146 | 0.0296 | 0.4527 | 0.5217 | 0.2129 | 0.2236 |
| RoPINNs | 0.1650 | 0.1720 | 0.0070 | 0.0170 | 0.6350 | 0.7200 | 0.1545 | 0.1622 |
| KAN | 0.1433 | 0.1458 | 0.0166 | 0.0343 | 0.6049 | 0.6587 | 0.0901 | 0.1042 |
| PINNMamba | 0.0197 | 0.0199 | 0.0094 | 0.0217 | 0.0188 | 0.0201 | 0.0535 | 0.0583 |
| PINN_SOAP | 0.2825 | 0.2851 | 0.0048 | 0.0096 | 0.0340 | 0.0363 | 0.0098 | 0.0086 |
| PirateNet | 0.2544 | 0.2637 | 0.0589 | 0.0965 | 0.9704 | 0.9704 | **0.0005** | **0.0005** |
| **ResPINN (ours)** | **0.0130** | **0.0154** | **0.0047** | **0.0075** | **0.0028** | **0.0046** | 0.0035 | 0.0048 |

operate without subsequencing. For the *PINNAcle* benchmark, dataset sizes and sampling details are provided in Appendix F. Residual flow blocks use a hidden dimension of 64, with three fully connected layers per block followed by a skip connection. The stagewise variant begins with three blocks and adds two new blocks at each stage, freezing the earlier ones. Neural ODE variants integrate a single residual block parameterization with a 4th-Order Runge-Kutta (RK4) solver (See Appendix G).

### 5.1. Do Residual flows mitigate Failure Modes?

We first benchmark *ResPINN* against recent PINN variants. Table 2 reports relative mean absolute error (rMAE) and relative root mean squared error (rRMSE) (See Appendix E for more details about the metrics). Classical PINNs perform poorly on Reaction and Convection, consistent with known failure modes. Both *PINNsFormer* and *PINNMamba* incorporate residual connections, but only at the level of one or two residual blocks. In contrast, *ResPINN* stacks residual updates throughout the network, directly instantiating the residual flow formulation. Across all four PDEs, ResPINN achieves the lowest errors, often by an order of magnitude, showing that residual flows provide consistent improvements beyond the shallow residual structures of prior models. Qualitative comparisons in Figure 3 confirm this pattern: Models with residual pathways achieve constructive reconstructions whereas vanilla PINNs suffer a larger deviation.

### 5.2. Iterative Refinement and Gradient Alignment

We next ask whether residual flows in PINNs act primarily as feature learners or as iterative refiners of predictions. We investigate this from two complementary perspectives.

For each block $T_i$, we measure the relative update size $\frac{\|T_i(z_i)\|}{\|z_i\|}$ averaged across sample points. Large values indicate substantial representation change (feature learning), while small values indicate incremental corrections (refinement). Figure 2 shows that in standard PINNs the ratio remains large across depth, whereas in ResPINNs it decreases steadily, consistent with refinement dynamics. For details on other PDEs, see Appendix I.

To probe whether the individual blocks can contribute to failure modes, we adopt the progressive-flow setting. At each training stage, after adding new residual blocks and freezing earlier ones, we train only a linear projection head to read out predictions from intermediate stages. Figure 6 in Appendix I illustrates that early stages incur high error similar to failure modes, but later stages systematically reduce error while leaving earlier predictions unchanged. This confirms that new blocks act as refiners rather than relearners, mirroring multistage correction in classical solvers.

### 5.3. Ablation Study

To disentangle the effect of discretization from architectural or activation choices, we compare the continuous-depth formulation (*O-PINN*, integrated with a fixed-RK4 ODE solver) against its discrete counterpart (*ResPINN*), each trained with either tanh or wavelet activations. This ablation allows us to test whether the improvements stem from the residual flow discretization itself or from particular activation functions. The results in Table 3 show that O-PINN and ResPINN exhibit complementary strengths: the continuous formulation benefits some PDE families (especially with wavelet activations), while discrete residual stacks remain competitive elsewhere.

### 5.4. Experiments on Complex Problems

To assess generalization, we evaluate on *PINNacle* (Zhongkai et al., 2024). On challenging multiscale tasks, baselines such as PINNsFormer (Zhao et al., 2024) and PINNMamba (Xu et al., 2025a) either fail to converge or

*Table 3.* Ablation on activation functions for continuous (O-PINN) and discrete (ResPINN) residual flow models. Results are reported on Wave, Reaction, and Convection PDEs using relative rMAE and rRMSE.

| Model | Wave | | Reaction | | Convection | |
|---|---|---|---|---|---|---|
| | rMAE | rRMSE | rMAE | rRMSE | rMAE | rRMSE |
| O-PINN + $\tanh$ | 0.038 | 0.039 | 0.018 | 0.035 | 0.014 | 0.016 |
| O-PINN + wavelet | 0.053 | 0.059 | 0.003 | 0.005 | 0.003 | 0.003 |
| ResPINN + $\tanh$ | 0.030 | 0.030 | 0.008 | 0.017 | 0.015 | 0.016 |
| ResPINN + wavelet | 0.070 | 0.074 | 0.008 | 0.009 | 0.006 | 0.006 |

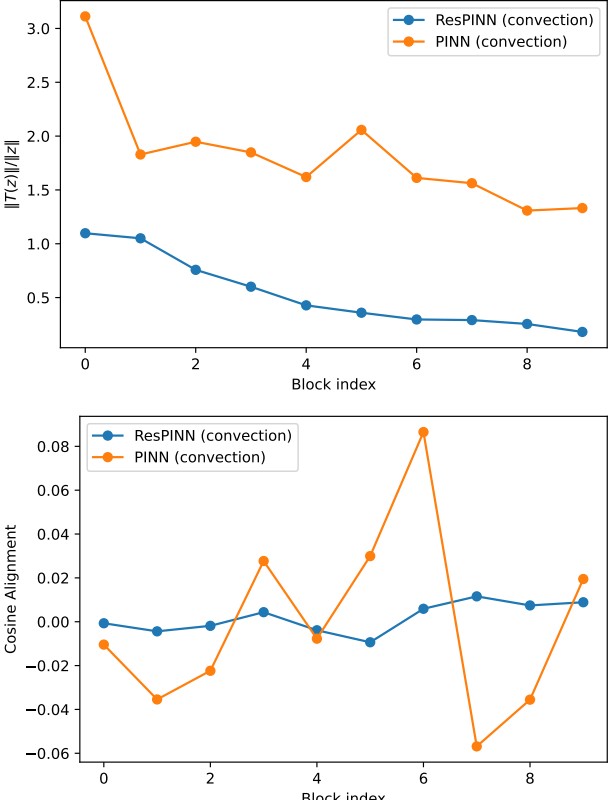

*Figure 2.* Left: relative update size $\|T_i(z_i)\|/\|z_i\|$ across depth. ResPINNs produce progressively smaller corrections, consistent with refinement. Right: Gradient Alignment. ResPINNs exhibit an almost neutral alignment with the local gradient descent.

run into out-of-memory errors, whereas *ResPINN* maintains comparable accuracy. We remark that the use of a full-batch L-BFGS can be challenging for chaotic PDEs which often require many collocation samples and may therefore run into memory limitations, as observed for some baselines. Details of the PINNacle experiments are shown in Appendix F. In addition, to further evaluate robustness on harder PDEs beyond the PINNacle suite, we include results on five additional equations: Allen–Cahn, Korteweg–de Vries, Ginzburg–Landau, lid-driven cavity, and Rayleigh–Taylor instability, comparing *ResPINN* with

PirateNet (Wang et al., 2024) both trained with an Adam optimizer. As reported in Table 6 (Appendix F), ResPINN achieves comparable or lower $L^2$ error across all cases, demonstrating stable performance even under complex non-linear and multiscale dynamics.

## 6. Conclusion

We reframed PINNs as *residual flows*: networks that solve PDEs by iteratively refining predictions through small residual updates. This view makes two optimzation effects explicit—*gradient alignment* (updates aligned with descent) and *Jacobian neutrality* (near-identity per-step Jacobians)—and led to simple instantiations (ResPINN, O-PINN, progressive residual flows).

Across canonical PDEs and the PINNacle suite, ResPINN achieved consistently lower errors. Mechanistic diagnostics support our hypotheses: residual blocks operate in the small-step regime (H1), maintain near-identity Jacobians across depth (H2), and exhibit iterative refinement (H3) as shown by decreasing update ratios in ResPINNs and stagewise error telescoping in curriculum training. These gains persist across activations, and the continuous formulation (O-PINN) can be advantageous on some PDE families, suggesting that continuous-time parameterizations merit exploration for very deep regimes.

Future work may explore how different numerical solvers induce distinct refinement behaviors, and whether ODE parameterizations applied directly in solution space for time-dependent PDEs can further mitigate failure modes. Bridging local theoretical insights with global behaviors observed in practice offers a promising avenue for deepening our understanding of residual flows.

## Acknowledgements

The authors acknowledge support by the DFG through FOR 5359 (ID 459419731), TRR 375 (ID 511263698), SPP 2298 (IDs 441826958 and 441826958), and SPP 2331 (IDs 441958259, 553345933, and 466468799), by the Carl-Zeiss Foundation through the initiative AI-Care, and by the BMFTR award 01IS24071A.

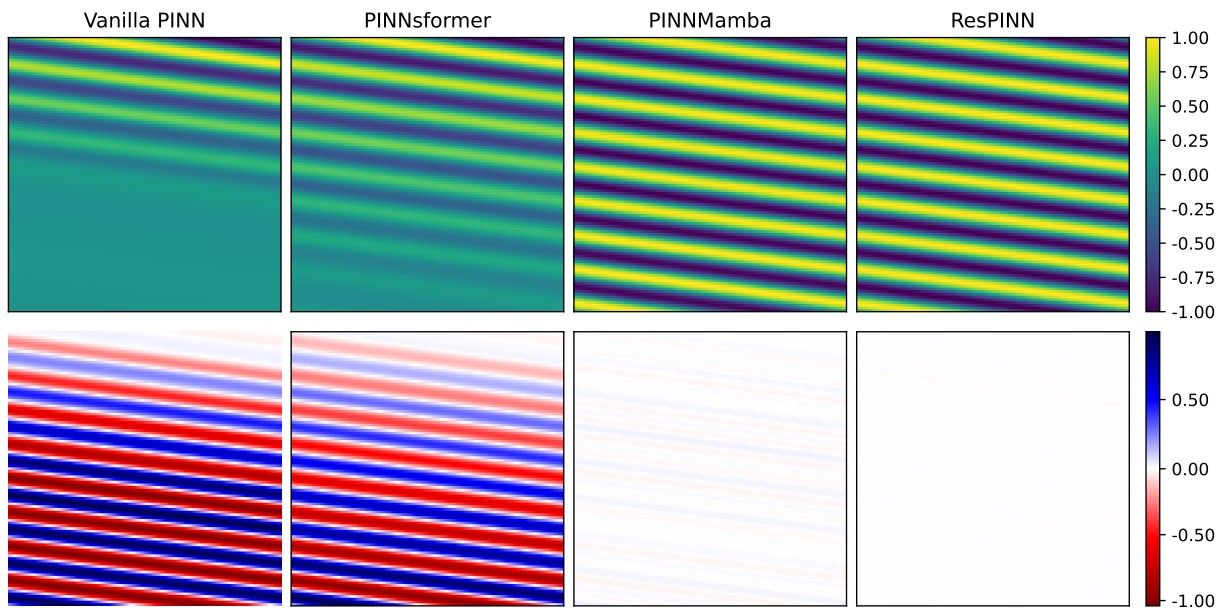

*Figure 3.* Qualitative comparison on Convection PDE. Top: predicted solutions. Bottom: pointwise errors.

## Impact Statement

This paper improves the robustness of Physics-Informed Neural Networks (PINNs) for solving PDEs by reformulating them as residual flows that iteratively refine predictions. More stable training can make physics-informed surrogates more practical for simulation and analysis in science and engineering (e.g., fluids, waves, diffusion, and related dynamical systems), potentially reducing compute costs and improving accuracy.

These improvements may broaden the practical applicability of PINNs in scientific computing and engineering, including surrogate modeling, inverse problems, and data-efficient simulation where enforcing known physical structure is beneficial. We do not foresee direct negative societal impacts arising uniquely from this contribution beyond those commonly associated with general improvements in machine-learning-based modeling tools.

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

## A. Reproducibility Statement.

All PDE setups (governing equations, domains, analytic solutions, and meshes) are detailed in Appendix E. Theoretical results and proofs appear in Appendix C, with the mean-field shattering adaptation in Appendix D. Architectural and solver specifications for ResPINN, O-PINN, and Progressive Flow are given in Appendix G, and additional alignment/refinement diagnostics are in Appendix I. PINNacle task definitions and results are reported in Appendix F. An anonymous code repository containing implementations of residual flows and scripts reproducing the experiments is available at `https://anonymous.4open.science/r/resflows-0FD5`

## B. The Use of Large Language Models (LLMs)

We used large language models solely for surface-level editing: spelling and grammar correction, and minor wording improvements. LLMs were *not* used for idea generation, experiment design, data analysis, coding, mathematical derivations, or substantive content creation.

## C. Proofs

We study feature evolution through a latent flow induced by residual transformations in continuous network time:

$$\frac{dz(k)}{dk} = T\big(z(k), k; x, t\big), \qquad k \in [0, K], \qquad z(0) = z_0 := E(x, t) \in \mathbb{R}^{d_h}. \tag{7}$$

*Remark* C.1 (Analytical surrogate). Equation equation 7 is not intended as the literal dynamics of fully connected PINNs, but as an analytical surrogate that lets us study feature evolution and gradient misalignment using the language of residual flows.

**Lemma C.2** (Integral form). *If $T(\cdot, \cdot)$ is continuous, then $z$ is a solution of equation 7 on $[0, K]$ if and only if*

$$z(k) = z_0 + \int_0^k T\big(z(\tau), \tau; x, t\big) \, d\tau, \qquad 0 \le k \le K. \tag{8}$$

*Proof.* ($\Rightarrow$) *Integrate equation 7 from 0 to $k$ to obtain equation 8.*
($\Leftarrow$) *If equation 8 holds and $T(z(\tau), \tau)$ is continuous in $\tau$, then by the fundamental theorem of calculus the map $k \mapsto z(k)$ is differentiable with $\frac{dz}{dk} = T(z(k), k)$ and $z(0) = z_0$, i.e., $z$ solves equation 7.* $\square$

**Theorem C.3** (Banach contraction mapping). *Let $(X, \| \cdot \|_X)$ be a Banach space and let $F : X \to X$ satisfy*

$$\|F(z) - F(z')\|_X \le c \, \|z - z'\|_X, \qquad \forall z, z' \in X,$$

*for some $0 < c < 1$. Then $F$ admits a unique fixed point $z^* \in X$, and the iterates $z^{(n+1)} = F(z^{(n)})$ converge to $z^*$ for any initial $z^{(0)} \in X$.*

**Theorem C.4** (Existence and uniqueness of a solution of a residual flow). *Let $T : \mathbb{R}^{d_h} \times [0, K] \to \mathbb{R}^{d_h}$ be continuous and assume there exists $L > 0$ such that*

$$\|T(z_1, k) - T(z_2, k)\| \le L \, \|z_1 - z_2\|, \qquad \|T(z, k)\| \le L(1 + \|z\|), \tag{9}$$

*for all $z, z_1, z_2 \in \mathbb{R}^{d_h}$ and $k \in [0, K]$. Then the IVP 7 admits a unique solution $z \in C([0, K], \mathbb{R}^{d_h})$.*

*Proof.* Fix $\delta > 0$ and consider the Banach space $X = C([0, \delta], \mathbb{R}^{d_h})$ with norm $\|z\|_X = \sup_{0 \le s \le \delta} \|z(s)\|$. Define the flow operator

$$(\mathcal{F}z)(k) := z_0 + \int_0^k T(z(\tau), \tau) \, d\tau.$$

If $\mathcal{F}z = z$, then by Lemma C.2, $z$ solves the IVP on $[0, \delta]$.

For $z, z' \in X$ and $k \in [0, \delta]$,

$$\|(\mathcal{F}z)(k) - (\mathcal{F}z')(k)\| \le \int_0^k \|T(z(\tau), \tau) - T(z'(\tau), \tau)\| \, d\tau \le L\delta \|z - z'\|_X.$$

Thus $\|\mathcal{F}z - \mathcal{F}z'\|_X \leq L\delta\|z - z'\|_X$, so $\mathcal{F}$ is a contraction whenever $L\delta < 1$. By Theorem C.3, $\mathcal{F}$ has a unique fixed point in $X$, which is the unique solution on $[0, \delta]$. Repeating the argument on successive intervals of length $\delta$ extends the solution uniquely to all of $[0, K]$. $\qquad\square$

**Definition C.5** (Discrete Residual Step). Let $\Delta k > 0$ and $k_n := n\,\Delta k$ for $n = 0, \ldots, N$ with $N\Delta k = K$. The explicit Euler discretization of the residual flow $\frac{dz(k)}{dk} = T(z(k), k; x, t)$ with $z(0) = z_0 := E(x, t)$ is

$$z_{n+1} \;=\; z_n + \Delta k\, T(z_n, k_n; x, t), \qquad z_0 = E(x, t).$$

Equivalently, this is a residual update with $T_n(z_n) := \Delta k\, T(z_n, k_n; x, t)$.

**Definition C.6** (Convergence/order). Let $z(\cdot)$ denote the (unique) solution of the IVP on $[0, K]$. A time-stepping scheme producing $\{z_n\}_{n=0}^N$ is said to *converge with order $p$* on $[0, K]$ if there exists a constant $C$, independent of $\Delta k$, such that

$$\max_{0 \leq n \leq N} \big\|z(k_n) - z_n\big\| \;\leq\; C\,(\Delta k)^p.$$

**Theorem C.7** (First-order convergence of the residual flows). *Assume the hypotheses of existence/uniqueness hold (global Lipschitz and linear growth in $z$ for $T$), and that the solution $z$ is twice continuously differentiable on $[0, K]$. Let $\{z_n\}$ be defined by C.5. Then the discrete formulation of the residual flows converges with order 1:*

$$\max_{0 \leq n \leq N} \big\|z(k_n) - z_n\big\| \;\leq\; C_K\,\Delta k,$$

*where $C_K$ depends on $K$, the Lipschitz constant $L$ of $T$ in $z$, and $\max_{k \in [0,K]} \|\ddot{z}(k)\|$, but is independent of $\Delta k$. Sketch. Taylor expand $z(k_{n+1}) = z(k_n) + \Delta k\,\dot{z}(k_n) + R_n$ with $\|R_n\| \leq C\,(\Delta k)^2$. Using $\dot{z}(k_n) = T(z(k_n), k_n)$ and subtracting the Euler step gives the error recurrence $e_{n+1} \leq (1 + L\Delta k)\,e_n + C\,(\Delta k)^2$, where $e_n := \|z(k_n) - z_n\|$. Apply the discrete Grönwall lemma to obtain $e_n \leq C\,\frac{e^{Lk_n}-1}{L}\,\Delta k \leq C\,\frac{e^{LK}-1}{L}\,\Delta k$.*

**Proposition C.8** (Gradient alignment in residual flows). *Let $\mathcal{L} : \mathbb{R}^{d_h} \to \mathbb{R}$ be continuously differentiable, and let $z : [0, K] \to \mathbb{R}^{d_h}$ be a continuously differentiable solution of the residual flow IVP equation 7. Then, for all $k \in [0, K]$,*

$$\frac{d}{dk}\,\mathcal{L}\big(z(k)\big) \;=\; \big\langle \nabla\mathcal{L}\big(z(k)\big),\, T\big(z(k), k; x, t\big)\big\rangle. \tag{10}$$

*1. If*

$$\big\langle \nabla\mathcal{L}(z(k)),\, T(z(k), k)\big\rangle \;\leq\; 0 \quad \textit{for all } k \in [0, K], \tag{11}$$

*then $\mathcal{L}(z(k))$ is nonincreasing on $[0, K]$.*

*2. If there exists a constant $c \in (0, 1]$ such that*

$$\frac{\big\langle T(z(k), k),\, -\nabla\mathcal{L}(z(k))\big\rangle}{\|T(z(k), k)\|\,\|\nabla\mathcal{L}(z(k))\|} \;\geq\; c \quad \textit{and} \quad \|T(z(k), k)\| > 0 \quad \textit{for all } k \in I \subset [0, K], \tag{12}$$

*then $\mathcal{L}(z(k))$ is strictly decreasing on $I$.*

*Proof.* The chain rule gives equation 10. Under equation 11, $\frac{d}{dk}\mathcal{L}(z(k)) \leq 0$ for all $k$, so $\mathcal{L}(z(k))$ is nonincreasing.

For equation 12, write

$$\frac{d}{dk}\,\mathcal{L}(z(k)) = -\,\|\nabla\mathcal{L}(z(k))\|\,\|T(z(k), k)\|\,\frac{\big\langle T(z(k), k),\, -\nabla\mathcal{L}(z(k))\big\rangle}{\|T(z(k), k)\|\,\|\nabla\mathcal{L}(z(k))\|} \tag{13}$$

$$\leq\; -c\,\|\nabla\mathcal{L}(z(k))\|\,\|T(z(k), k)\|. \tag{14}$$

On any interval $I$ where $c > 0$ and $\|T(z(k), k)\| > 0$, the right-hand side is strictly negative, hence $\mathcal{L}(z(k))$ is strictly decreasing on $I$. $\qquad\square$

*Proof of Lemma 3.2.* Since $\mathcal{L}$ has $\beta$-Lipschitz continuous gradient, we have for any $u, v$ in a neighborhood of $z_k$:

$$\mathcal{L}(v) \leq \mathcal{L}(u) + \langle \nabla\mathcal{L}(u), v - u \rangle + \tfrac{\beta}{2}\|v - u\|^2.$$

Apply this inequality with $u = z_k$ and $v = z_{k+1} = z_k + T_k(z_k)$:

$$\mathcal{L}(z_{k+1}) \leq \mathcal{L}(z_k) + \langle \nabla_{z_k}\mathcal{L}(z_k), T_k(z_k) \rangle + \tfrac{\beta}{2}\|T_k(z_k)\|^2.$$

To capture the effect of depth, note that subsequent updates depend on how $T_j$ is transformed through the Jacobians $J_\ell = \partial z_{\ell+1}/\partial z_\ell$. The gradient at $z_k$ is related to the gradient at $z_{k+1}$ by the chain rule:

$$\nabla_{z_k}\mathcal{L}(z_{k+1}) = J_k^\top \nabla_{z_{k+1}}\mathcal{L}(z_{k+1}).$$

Rolling this back from step $K$ to step $k$ shows that each local Lipschitz constant is scaled by the squared operator norms of the Jacobians:

$$\|\nabla^2\mathcal{L}(z_k)\|_2 \ \leq \ \beta\Big(\prod_{\ell=k}^{K-1}\|J_\ell\|_2\Big)^2.$$

Therefore there exists a local smoothness constant $\beta_k \leq \beta\big(\prod_{\ell=k}^{K-1}\|J_\ell\|_2\big)^2$ such that

$$\mathcal{L}(z_{k+1}) \leq \mathcal{L}(z_k) + \langle \nabla_{z_k}\mathcal{L}(z_k), T_k(z_k) \rangle + \tfrac{\beta_k}{2}\|T_k(z_k)\|^2.$$

This completes the proof. $\qquad\square$

# D. Mean-field gradient shattering for PINN Jacobians

We use the mean-field perspective developed in prior studies of deep random networks (Poole et al., 2016; Balduzzi et al., 2017; Pennington et al., 2018; Yang & Schoenholz, 2017) to interpret the behavior of input-output Jacobians in PINNs. The following calculation records the infinite-width scaling prediction for Jacobian norms and cross-input Jacobian correlations.

**Theorem D.1** (Mean-field Jacobian decorrelation). *Consider a depth-$L$, width-$n$ fully-connected network*

$$x^0(z) = z, \qquad h^\ell(z) = W^\ell x^{\ell-1}(z), \qquad x^\ell(z) = \phi(h^\ell(z)), \qquad \ell = 1, \ldots, L,$$

*with $W_{ij}^\ell \sim \mathcal{N}(0, \sigma_w^2/n)$ independently, and scalar output*

$$u_\theta(z) = v^\top x^L(z), \qquad v_i \sim \mathcal{N}(0, \sigma_v^2/n),$$

*independently of the hidden weights. Let*

$$J_\theta(z) = \nabla_z u_\theta(z).$$

*Assume that the mean-field variance recursion has a finite positive fixed point $q_*$, and that for the pair of distinct inputs $z, z'$, the corresponding correlation recursion converges to a limit $c_* \in (-1, 1)$. Define*

$$\chi = \sigma_w^2 \mathbb{E}_{g \sim \mathcal{N}(0,q_*)}\big[\phi'(g)^2\big],$$

*and*

$$\chi_{12} = \sigma_w^2 \mathbb{E}_{(g,g') \sim \mathcal{N}(0,\Sigma_*)}\big[\phi'(g)\phi'(g')\big],$$

*where*

$$\Sigma_* = q_* \begin{pmatrix} 1 & c_* \\ c_* & 1 \end{pmatrix}.$$

*Then, in the infinite-width mean-field limit,*

$$\mathbb{E}\|J_\theta(z)\|_F^2 = \exp(o(L))\chi^L,$$

*and the normalized second-moment Jacobian correlation satisfies*

$$\frac{\mathbb{E}\langle J_\theta(z), J_\theta(z') \rangle}{\sqrt{\mathbb{E}\|J_\theta(z)\|_F^2 \mathbb{E}\|J_\theta(z')\|_F^2}} = \mathcal{O}\left(\left|\frac{\chi_{12}}{\chi}\right|^L\right).$$

*In particular, when*

$$\rho := \left| \frac{\chi_{12}}{\chi} \right| < 1,$$

*the normalized Jacobian covariance decays exponentially with depth. If, in addition, Jacobian norms concentrate around their mean-field second moments, this second-moment decorrelation is consistent with exponential decay of the empirical Jacobian cosine similarity. The case $\chi = 1$ corresponds to the mean-field edge-of-chaos condition for Jacobian norm preservation; when $\chi < 1$ the Jacobian norm decays exponentially, and when $\chi > 1$ it grows exponentially.*

*Proof.* For each layer, the mean-field limit gives deterministic recursions for the marginal variance and the cross-input covariance. Specifically, as $n \to \infty$, the preactivations $h_i^\ell(z)$ become i.i.d. Gaussian with variance

$$q^\ell = \mathbb{E}[h_i^\ell(z)^2],$$

and for two inputs $z, z'$, their correlation

$$c^\ell = \frac{\mathbb{E}[h_i^\ell(z)h_i^\ell(z')]}{q^\ell}$$

evolves according to the standard mean-field recursion

$$q^{\ell+1} = \sigma_w^2 \mathbb{E}_{g \sim \mathcal{N}(0,q^\ell)}[\phi(g)^2],$$

and

$$c^{\ell+1} = \frac{\sigma_w^2}{q^{\ell+1}} \mathbb{E}_{(g,g') \sim \mathcal{N}(0,\Sigma^\ell)}[\phi(g)\phi(g')],$$

where

$$\Sigma^\ell = q^\ell \begin{pmatrix} 1 & c^\ell \\ c^\ell & 1 \end{pmatrix}.$$

By assumption, $q^\ell \to q_*$ and $c^\ell \to c_* \in (-1, 1)$.

The input-output Jacobian is

$$J_\theta(z) = v^\top D^L(z) W^L D^{L-1}(z) W^{L-1} \cdots D^1(z) W^1,$$

where

$$D^\ell(z) = \mathrm{diag}(\phi'(h^\ell(z))).$$

Let

$$G^\ell(z) = D^\ell(z) W^\ell D^{\ell-1}(z) W^{\ell-1} \cdots D^1(z) W^1.$$

Thus

$$J_\theta(z) = v^\top G^L(z).$$

We first compute the growth of the second moment of the Jacobian norm. By independence, isotropy of the Gaussian weights, and the mean-field law of large numbers, each layer multiplies the squared norm of a typical input sensitivity by

$$\chi_\ell = \sigma_w^2 \mathbb{E}_{g \sim \mathcal{N}(0,q^\ell)}[\phi'(g)^2].$$

Hence

$$\mathbb{E}\|J_\theta(z)\|_F^2 = C_z \prod_{\ell=1}^L \chi_\ell,$$

where $C_z$ is independent of $L$. Since $q^\ell \to q_*$, we have

$$\chi_\ell \to \chi = \sigma_w^2 \mathbb{E}_{g \sim \mathcal{N}(0,q_*)}[\phi'(g)^2].$$

Therefore,

$$\mathbb{E}\|J_\theta(z)\|_F^2 = \Theta(\chi^L).$$

This proves the norm growth and decay claim.

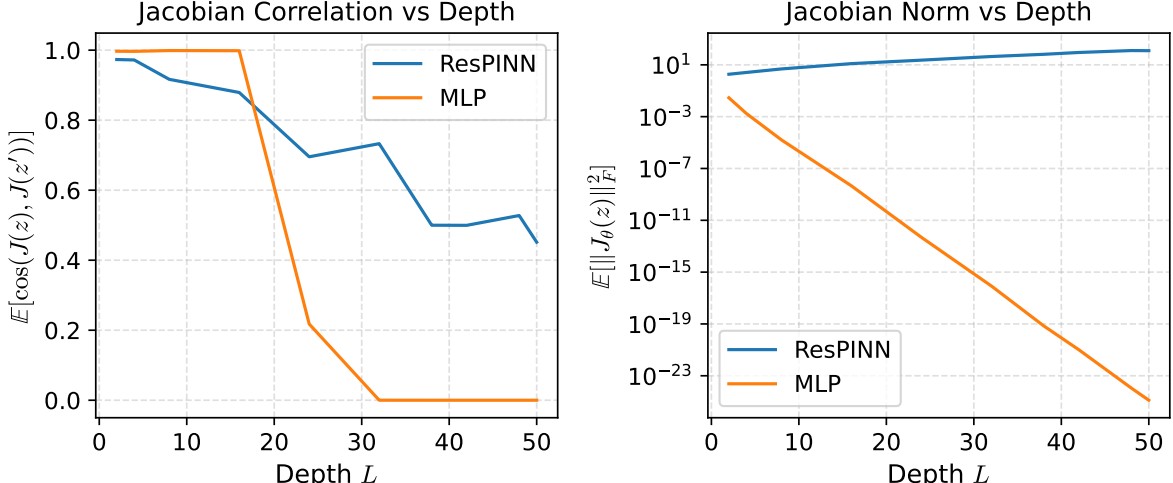

*Figure 4.* Left: Correlation between Jacobians. The cosine similarity of input-output Jacobians decay rapidly with depth for standard MLP PINNs. Residual PINNs maintain significantly higher gradient correlation across depth. Right: Jacobian Norms across depth(log scale). The expected Jacobian norm decreases exponentially with depth for MLP PINNs, while residual architectures maintain stable gradient magnitudes. Both plots were obtained from one common setup: fully connected and residual networks with tanh activation and width 128 were evaluated on a fixed grid of 256 points over the domain $(-1, 1) \times (0, 1)$, and all reported quantities represent averages over 20 independent random seeds.

Now consider two distinct inputs $z, z'$. The same layerwise computation applied to the joint backward sensitivities gives

$$\mathbb{E}\langle J_\theta(z), J_\theta(z')\rangle = C_{z,z'} \prod_{\ell=1}^{L} \chi_{12,\ell},$$

where

$$\chi_{12,\ell} = \sigma_w^2 \mathbb{E}_{(g,g')\sim\mathcal{N}(0,\Sigma^\ell)}[\phi'(g)\phi'(g')].$$

Since $\Sigma^\ell \to \Sigma_*$, we obtain

$$\chi_{12,\ell} \to \chi_{12} = \sigma_w^2 \mathbb{E}_{(g,g')\sim\mathcal{N}(0,\Sigma_*)}[\phi'(g)\phi'(g')].$$

Thus

$$\mathbb{E}\langle J_\theta(z), J_\theta(z')\rangle = \exp(o(L))\chi_{12}^L.$$

Combining this with

$$\mathbb{E}\|J_\theta(z)\|_F^2 = \exp(o(L))\chi^L, \qquad \mathbb{E}\|J_\theta(z')\|_F^2 = \exp(o(L))\chi^L,$$

gives

$$\frac{\mathbb{E}\langle J_\theta(z), J_\theta(z')\rangle}{\sqrt{\mathbb{E}\|J_\theta(z)\|_F^2 \mathbb{E}\|J_\theta(z')\|_F^2}} = \exp(o(L))\left(\frac{\chi_{12}}{\chi}\right)^L.$$

Therefore, if

$$\rho = \left|\frac{\chi_{12}}{\chi}\right| < 1,$$

the normalized second-moment Jacobian correlation decays exponentially with depth. Under an additional concentration assumption for the Jacobian norms, this second-moment decorrelation also predicts exponential decay of empirical Jacobian cosine similarity. □

**Empirical illustration.** The mean-field analysis predicts two depth-dependent effects at random initialization: decay of Jacobian correlations across nearby inputs and exponential growth or decay of Jacobian norms. Figure 4 provides an empirical check of these predictions by measuring input-output Jacobians for fully connected and residual PINNs of increasing depth. For standard MLP PINNs, the Jacobian cosine similarity decreases rapidly with depth, and the Jacobian norm collapses on a logarithmic scale, consistent with reduced input sensitivity. Residual PINNs show a weaker decay in correlation and preserve larger Jacobian norms across depth. These observations are consistent with the view that residual connections mitigate, rather than eliminate, the mean-field shattering mechanism.

# E. PDE Setups and Metrics

## E.1. Metrics

In our experiments, we report three metrics: the relative mean absolute error (rMAE), and the relative root mean squared error (rMSE). For a set of evaluation points $\mathcal{S}$, model prediction $u_\theta$, and ground-truth solution $u^*$, we define

$$\text{rMAE} = \frac{\sum_{x \in \mathcal{S}} \left| u_\theta(x) - u^*(x) \right|}{\sum_{x \in \mathcal{S}} \left| u^*(x) \right|}, \tag{15}$$

$$\text{rMSE} = \sqrt{\frac{\sum_{x \in \mathcal{S}} \left( u_\theta(x) - u^*(x) \right)^2}{\sum_{x \in \mathcal{S}} \left( u^*(x) \right)^2}}. \tag{16}$$

Note that both $u_\theta(x)$ and $u^*(x)$ can take positive or negative values; consequently, rMAE and rMSE may exceed 1.

## E.2. Benchmarks

To comprehensively test our algorithm, we include four benchmarks. The first three correspond to canonical PDEs widely used in the PINN literature, while the last one is the large-scale *PINNacle* benchmark (Zhongkai et al., 2024).

**1D–Reaction.** This one-dimensional nonlinear ODE models chemical reactions:

$$\frac{\partial u}{\partial t} - \rho u(1 - u) = 0, \quad x \in (0, 2\pi), \ t \in (0, 1),$$

with initial and boundary conditions

$$u(x, 0) = \exp\left( -\frac{(x - \pi)^2}{2(\pi/4)^2} \right), \quad u(0, t) = u(2\pi, t).$$

The analytic solution is

$$u(x, t) = \frac{h(x)e^{\rho t}}{h(x)e^{\rho t} + 1 - h(x)}, \quad h(x) = \exp\left( -\frac{(x - \pi)^2}{2(\pi/4)^2} \right),$$

with $\rho = 5$. Prior work (Raissi et al., 2019; Krishnapriyan et al., 2021) identified this case as a "PINN failure mode" due to the nonlinear term, and its sharp interior boundary adds further difficulty. Following PINNsFormer (Xu et al., 2025a), we sample 101 points on the initial/boundary sets and a $101 \times 101$ grid on the residual domain. Evaluation uses the same mesh.

**1D–Wave.** A standard hyperbolic PDE from acoustics and fluid dynamics:

$$\frac{\partial^2 u}{\partial t^2} - 4\frac{\partial^2 u}{\partial x^2} = 0, \quad x \in (0, 1), \ t \in (0, 1),$$

with initial and boundary conditions

$$u(x, 0) = \sin(\pi x) + \tfrac{1}{2}\sin(\beta \pi x), \quad \frac{\partial u(x, 0)}{\partial t} = 0, \quad u(0, t) = u(1, t) = 0.$$

*Table 4.* PDE benchmarks from PINNacle (Zhongkai et al., 2024). We list input dimensionality, training/testing sizes, and representative simplified equations. All PDEs here are second-order. Full formalizations, coefficient meanings, and boundary/initial conditions appear in (Zhongkai et al., 2024).

| PDE | Dimension | $N_{\text{train}}$ | $N_{\text{test}}$ | Key Equation |
|---|---|---|---|---|
| Burgers | 1D+Time (1d-C) | 16384 | 12288 | $\frac{\partial u}{\partial t} + u \cdot \nabla u - \nu \Delta u = 0$ |
| | 2D+Time (2d-C) | 98308 | 82690 | same form in 2D |
| Poisson | 2D (2d-C) | 12288 | 10240 | $-\Delta u = 0$ |
| | 2D (2d-CG) | 12288 | 10240 | $-\Delta u + k^2 u = f(x,y)$ |
| | 3D (3d-CG) | 49152 | 40960 | $-\mu_i \Delta u + k_i^2 u = f(x,y,z), \ i = 1,2$ |
| | 2D (2d-MS) | 12288 | 10329 | $-\nabla(a(x)\nabla u) = f(x,y)$ |
| Heat | 2D+Time (2d-VC) | 65536 | 49189 | $\frac{\partial u}{\partial t} - \nabla(a(x)\nabla u) = f(x,t)$ |
| | 2D+Time (2d-MS) | 65536 | 49189 | $\frac{\partial u}{\partial t} - \frac{1}{(500\pi)^2}u_{xx} - \frac{1}{\pi^2}u_{yy} = 0$ |
| | 2D+Time (2d-CG) | 65536 | 49152 | $\frac{\partial u}{\partial t} - \Delta u = 0$ |
| Navier–Stokes | 2D (2d-C) | 14337 | 12378 | $u \cdot \nabla u + \nabla p - \frac{1}{Re}\Delta u = 0, \ \nabla \cdot u = 0$ |
| | 2D (2d-CG) | 14055 | 12007 | same form |
| Wave | 1D+Time (1d-C) | 12288 | 10329 | $u_{tt} - 4u_{xx} = 0$ |
| | 2D+Time (2d-CG) | 49170 | 42194 | $\left[\nabla^2 - \frac{1}{c(x)}\frac{\partial^2}{\partial t^2}\right]u(x,t) = 0$ |
| Chaotic (GS) | 2D+Time | 65536 | 61780 | $\begin{cases} u_t = \varepsilon_1 \Delta u + b(1-u) - uv^2, \\ v_t = \varepsilon_2 \Delta v - dv + uv^2 \end{cases}$ |
| High-dim | 5D (P-Nd) | 49152 | 67241 | $-\Delta u = \frac{\pi^2}{4}\sum_{i=1}^{n}\sin\left(\frac{\pi}{2}x_i\right)$ |
| | 5D+Time (H-Nd) | 65537 | 49152 | $\frac{\partial u}{\partial t} = k\Delta u + f(x,t)$ |

The analytic solution is

$$u(x,t) = \sin(\pi x)\cos(2\pi t) + \tfrac{1}{2}\sin(\beta\pi x)\cos(2\beta\pi t),$$

with $\beta = 3$. Compared to Reaction and Convection, the solution is smoother, making it easier for deep models. Training/evaluation meshes are sampled as in Reaction.

**1D–Convection.** A hyperbolic PDE relevant in fluids, atmosphere, and heat transfer:

$$\frac{\partial u}{\partial t} + \beta\frac{\partial u}{\partial x} = 0, \quad x \in (0, 2\pi), \ t \in (0,1),$$

with

$$u(x,0) = \sin(x), \quad u(0,t) = u(2\pi, t).$$

The analytic solution is $u(x,t) = \sin(x - \beta t)$, where we set $\beta = 50$. Despite its simple closed form, this problem is challenging for PINNs due to the high-frequency oscillations and sharp loss landscape (Krishnapriyan et al., 2021). Training/evaluation meshes follow the same setup as above.

**Heat Equation:** The heat equation is a second-order parabolic PDE that describes heat distribution in a given region over time. It is a classic example of a diffusive system, which presents challenges related to numerical stiffness.

- **Equation**: $\frac{\partial u}{\partial t} = \alpha\frac{\partial^2 u}{\partial x^2}$, with $\alpha = 0.1$.

- **Domain**: $(x,t) \in [0,1] \times [0,1]$.

*Table 5.* Results on PINNacle. Baseline results are from Wu et al. (2024); Xu et al. (2025a). OOM means Out-of-Memory.

| | PINN | | PINNsFormer | | PINNMamba | | ResPINN | |
|---|---|---|---|---|---|---|---|---|
| Equation | rMAE | rRMSE | rMAE | rRMSE | rMAE | rRMSE | rMAE | rRMSE |
| Burgers 1d-C | 1.1e-2 | 3.3e-2 | 9.3e-3 | 1.4e-2 | 3.7e-3 | 1.1e-3 | 4.6e-3 | 1.4e-3 |
| Burgers 2d-C | 4.5e-1 | 5.2e-1 | OOM | OOM | OOM | OOM | OOM | OOM |
| Poisson 2d-C | 7.5e-1 | 6.8e-1 | 7.2e-1 | 6.6e-1 | 6.2e-1 | 5.7e-1 | 7.8e-1 | 7.1e-1 |
| Poisson 2d-CG | 5.4e-1 | 6.6e-1 | 5.4e-1 | 6.3e-1 | 1.2e-1 | 1.4e-1 | 4.4e-3 | 8.6e-3 |
| Poisson 3d-CG | 4.2e-1 | 5.0e-1 | OOM | OOM | OOM | OOM | OOM | OOM |
| Poisson 2d-MS | 7.8e-1 | 6.4e-1 | 1.3e+0 | 1.1e+0 | 7.2e-1 | 6.0e-1 | 9.0e-1 | 7.5e-1 |
| Heat 2d-VC | 1.2e+0 | 9.8e-1 | OOM | OOM | OOM | OOM | OOM | OOM |
| Heat 2d-MS | 4.7e-2 | 6.9e-2 | OOM | OOM | OOM | OOM | 6.5e-3 | 4.5e-3 |
| Heat 2d-CG | 2.7e-2 | 2.3e-2 | OOM | OOM | OOM | OOM | OOM | OOM |
| NS 2d-C | 6.1e-2 | 5.1e-2 | OOM | OOM | OOM | OOM | OOM | OOM |
| NS 2d-CG | 1.8e-1 | 1.1e-1 | 1.0e-1 | 7.0e-2 | 1.1e-2 | 7.8e-3 | 1.4e-2 | 9.8e-3 |
| Wave 1d-C | 5.5e-1 | 5.5e-1 | 5.0e-1 | 5.1e-1 | 1.0e-1 | 1.0e-1 | 3.4-2 | 3.7e-2 |
| Wave 2d-CG | 2.3e+0 | 1.6e+0 | OOM | OOM | OOM | OOM | OOM | OOM |
| Chaotic GS | 2.1e-2 | 9.4e-2 | OOM | OOM | OOM | OOM | OOM | OOM |
| High-dim PNd | 1.2e-3 | 1.1e-3 | OOM | OOM | OOM | OOM | OOM | OOM |
| High-dim HNd | 1.2e-2 | 5.3e-3 | OOM | OOM | OOM | OOM | OOM | OOM |

- **Initial Condition**: $u(x, 0) = \sin(\pi x)$.

- **Boundary Conditions**: $u(0, t) = 0$ and $u(1, t) = 0$ (Dirichlet).

- **Analytical Solution**: $u(x, t) = \sin(\pi x)e^{-\alpha \pi^2 t}$.

**PINNacle.** The fourth benchmark is *PINNacle* (Zhongkai et al., 2024), built on DeepXDE (Lu et al., 2021b). It comprises 16 PDE tasks covering fluid dynamics, heat conduction, nonlinear and multiscale phenomena, and high-dimensional settings. For a fair comparison with prior works (Wu et al., 2024; Xu et al., 2025a), we test on 7 problems as the combination with LBFGS and deep archictures can lead to OOM even on advanced GPU archtectures as previously noted by Xu et al. (2025a). Dataset details are summarized in Table 4.

# F. PINNacle PDE Benchmark and Harder PDEs

*Table 6.* Comparison of ResPINN and PirateNet on complex PDEs. Reported values are $L^2$ errors over the full spatio-temporal domain. Both models were trained and evaluated under identical settings. ResPINN attains comparable or lower error across most benchmarks, confirming that its stability extends to complex nonlinear PDEs.

| PDE | PirateNet | ResPINN |
|---|---|---|
| Allen–Cahn | $2.24 \times 10^{-5}$ | $\mathbf{2.19 \times 10^{-5}}$ |
| Korteweg–de Vries | $7.04 \times 10^{-4}$ | $\mathbf{5.05 \times 10^{-4}}$ |
| Ginzburg–Landau | $1.49 \times 10^{-2}$ | $\mathbf{4.01 \times 10^{-3}}$ |
| Lid-driven cavity (Re = $5 \times 10^3$) | $\mathbf{3.24 \times 10^{-1}}$ | $3.69 \times 10^{-1}$ |
| Rayleigh–Taylor instab. (Ra = $10^6$) | $\mathbf{7.32 \times 10^{-2}}$ | $9.63 \times 10^{-2}$ |

In addition to the PINNacle results in Table 5, we evaluate ResPINN and PirateNet on several harder PDEs: Allen–Cahn, Korteweg–de Vries, Ginzburg–Landau, lid-driven cavity, and Rayleigh–Taylor instability. Both models are trained under identical conditions, and accuracy is measured using the integrated $L^2$ error over the full spatio-temporal domain. As shown in Table 6, ResPINN attains comparable or lower error across all equations, demonstrating stable performance even on more complex, nonlinear dynamics. We note that, following prior works, we train with L-BFGS which is a full batch optimizer. In complex and high-dimensional problem, OOM issues can be mitigated by optimizers which use minibatches as we show in Table 6.

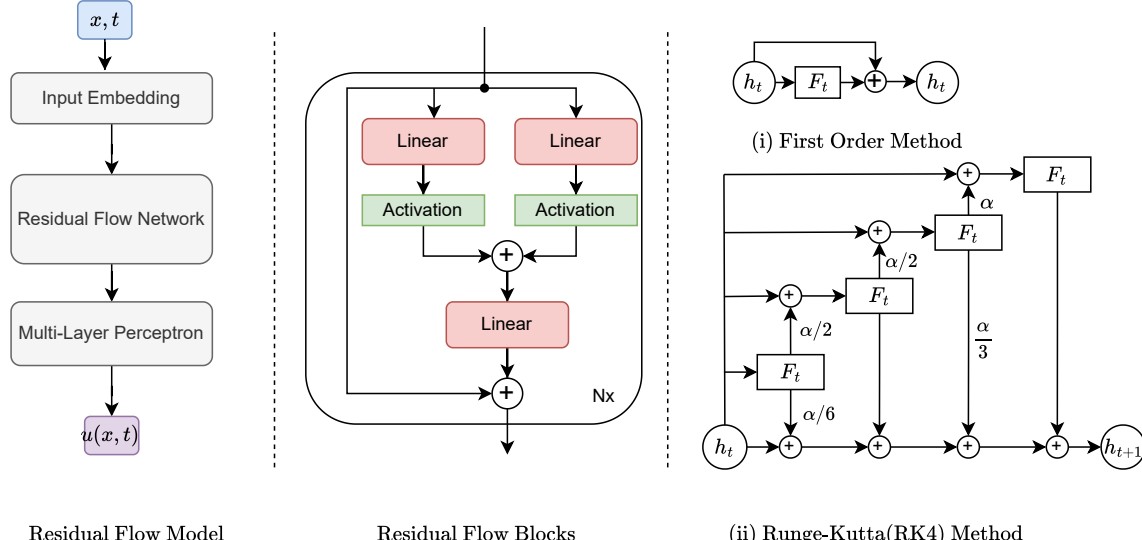

Residual Flow Model      Residual Flow Blocks      (ii) Runge-Kutta(RK4) Method

*Figure 5.* ResPINN overview. Inputs $(x, t)$ are encoded to a latent state $h(0)$, which is iteratively refined by a residual flow in pseudo-time $s$. The flow is realized either as a stacked residual (Euler) network or as a higher-order explicit solver RK4. The terminal state $h(S)$ is decoded to the PDE solution $u(x, t)$.

## G. ODE Solvers and Residual Flows

For completeness, we recall the connection between residual updates and classical numerical ODE solvers. Consider an ODE

$$\frac{dh(t)}{dt} = f(h(t), t), \qquad h(0) = h_0.$$

### G.1. Residual Flow Solvers

**Forward Euler.** The simplest explicit solver advances in steps of size $\alpha > 0$ via

$$h_{k+1} = h_k + \alpha\, f(h_k, t_k)$$

This is precisely the form of a residual block: each step applies a correction around the identity.

**Runge–Kutta (RK4).** Higher-order solvers reduce truncation error by evaluating $f$ at intermediate points. The classical fourth-order Runge–Kutta scheme computes

$$
\begin{aligned}
k_1 &= f(h_k, t_k), \\
k_2 &= f(h_k + \tfrac{\alpha}{2} k_1, t_k + \tfrac{\alpha}{2}), \\
k_3 &= f(h_k + \tfrac{\alpha}{2} k_2, t_k + \tfrac{\alpha}{2}), \\
k_4 &= f(h_k + \alpha k_3, t_k + \alpha), \\
h_{k+1} &= h_k + \tfrac{\alpha}{6}(k_1 + 2k_2 + 2k_3 + k_4).
\end{aligned}
$$

*ResPINNs* correspond to Euler-like discrete updates , while *O-PINNs* instantiate the continuous limit using RK4 integration with weight sharing. Implementattion details follow.

### G.2. Implementation of Residual Flows

**ResPINN (discrete residual stack).** A fixed-depth network composed of $K = 10$ residual blocks, each block containing three fully connected layers of width $64$ with a skip connection. A linear encoder maps inputs to latent space, and a single fully connected output head maps back to the PDE solution.

| PDE | rMAE | | | rMSE | | |
|---|---|---|---|---|---|---|
| | PINN64 | ResPINN32 | ResPINN64 | PINN64 | ResPINN32 | ResPINN64 |
| Wave | 0.0080 | 0.0130 | **0.0069** | 0.0081 | 0.0154 | **0.0068** |
| Reaction | 0.0271 | **0.0047** | 0.0058 | 0.0502 | **0.0075** | 0.0120 |
| Convection | 0.0059 | **0.0028** | 0.0046 | 0.0072 | **0.0046** | 0.0050 |
| Heat | **0.0003** | 0.0035 | 0.0005 | **0.0003** | 0.0048 | 0.0005 |

*Table 7.* Sensitivity to arithmetic precision. FP64 improves accuracy in PINNs at the cost of roughly doubled memory consumption. ResPINN in FP32 reaches accuracy levels comparable to FP64 PINN and exceeds it in several PDEs. Combining ResPINN with FP64 can further enhance stability and accuracy, providing additional gains where higher numerical precision is beneficial.

| Method | Params | Time per epoch (s) | Time Over-head | Peak Memory (MiB) | Memory Over-head | rMAE | rMSE |
|---|---|---|---|---|---|---|---|
| PINN | 527k | 1.01 | ×1 | 2028 | ×1 | 0.4101 | 0.4141 |
| ResPINN(ours) | **53k** | 1.27 | **×1.26** | 2025 | **×1** | **0.0130** | **0.0154** |
| PINNsFormer | 454k | 6.97 | ×6.90 | 19170 | ×9.45 | 0.3559 | 0.3622 |
| PINNMamba | 286k | 6.77 | ×6.70 | 15574 | ×7.68 | 0.0197 | 0.0199 |
| PirateNet | 724k | 3.12 | ×3.09 | 7324 | ×3.61 | 0.2544 | 0.2637 |

*Table 8.* Training Overhead and Accuracy (Wave PDE) with best computational and memory overheads highlighted in bold. ResPINN maintains accuracy improvements while remaining computationally lightweight. Its per epoch wall time and peak memory remain close to the Vanilla PINN baseline and far below sequence type models and PirateNet. The method provides these gains with only 1.26× time overhead and no memory increase. Sequence models require 6 to 9× more computation and 7 to 9× more memory. PirateNet incurs 3 to 4× overhead.

**O-PINN (continuous residual flow).** Uses the same residual block as the vector field $f_\theta$, but instead of stacking layers explicitly, the dynamics are integrated with RK4. This yields a continuous-depth model whose trajectory corresponds to an effectively deeper residual flow.

**Progressive Flow.** Starts with three residual blocks and adds two new blocks at each training stage while freezing earlier ones. Both encoder and decoder are linear projections, ensuring that representational capacity resides in the blocks. At each stage, the final projection layer is re-initialized and trained as a predictor of the PDE solution, providing a direct probe of iterative refinement.

An overview of ResPINN and O-PINN archirectures is shown in Figure 5.

## H. Training Overhead and Sensitivity to Precision

Several recent PINN variants introduce substantial compute and memory overhead, making it unclear whether their gains stem from architectural changes or increased resource budgets. Since our focus is on structural failure modes, we report explicit overhead comparisons and sensitivity to numerical precision. Prior work such as (Xu et al., 2025b) shows that arithmetic precision can materially affect optimization, so we evaluate both FP32 and FP64. Table 8 shows that ResPINN stays lightweight: its per-epoch time is only 1.26× the FP32 PINN baseline, and its peak memory use is unchanged. By contrast, transformer- and state-space–based PINNs require 6–9× more computation and 7–9× more memory, while PirateNet incurs a 3–4× overhead due to FP64 operation and added components. Table 7 shows that although FP64 stabilizes standard PINNs, ResPINN in FP32 already matches or exceeds FP64 PINN accuracy on several PDEs, with FP64 offering further gains when higher precision is helpful. These results indicate that ResPINN's improvements arise from its residual-flow structure rather than numerical precision or increased compute.

## I. Additional alignment plots

This appendix provides additional diagnostics that illuminate how residual flows alter optimization dynamics in PINNs. Rather than focusing on aggregate error metrics, the figures below examine *how* predictions evolve across depth and *how* individual residual blocks behave in terms of update magnitude and gradient alignment. Together, these plots support

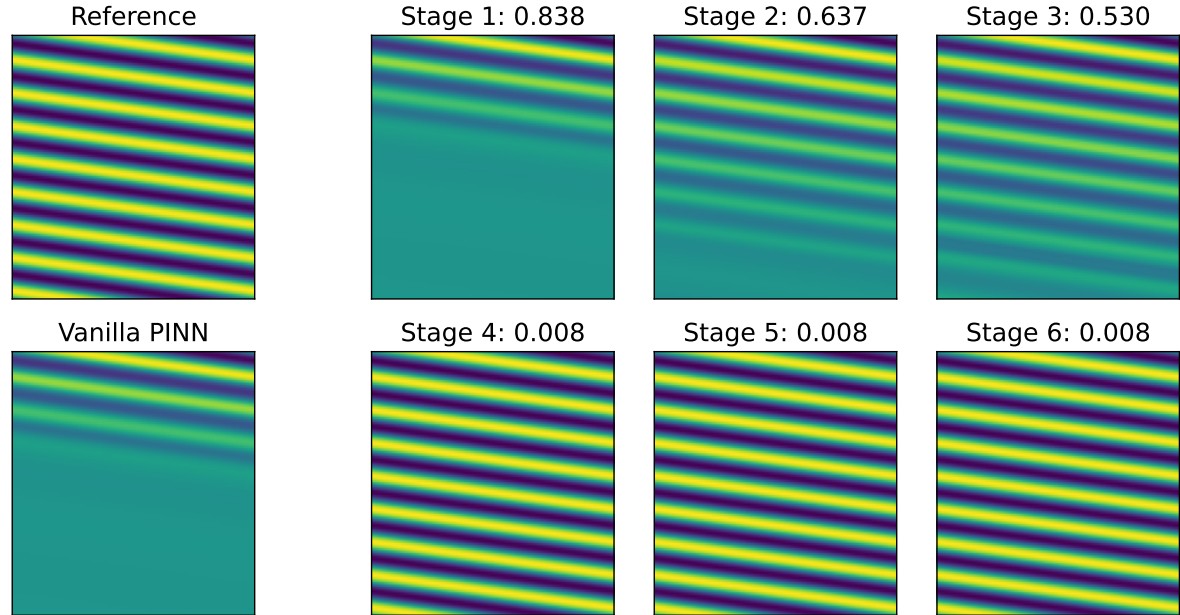

*Figure 6.* Predicted solutions across stages of the progressive residual flow procedure. The left column shows the reference prediction (top) and the vanilla PINN baseline (bottom) for the convection problem. The right panels present stagewise predictions annotated with stage index and rMSE. Early stages exhibit failure behavior and miss the temporal structure of the solution, while later stages progressively recover finer temporal dynamics.

the interpretation of residual flows as iterative refinement procedures that mitigate flow mismatch and stabilize gradient propagation.

**Stagewise refinement via progressive residual flows.** Figure 6 visualizes predictions across stages of the progressive residual flow procedure for the convection problem. The left column shows the reference solution and the vanilla PINN baseline, while the remaining panels show predictions after successive stages, annotated with rMSE. Early stages (Stages 1–2) clearly exhibit failure behavior: predictions are overly smooth and fail to capture the oscillatory temporal structure of the solution, closely resembling the vanilla PINN baseline. As additional residual blocks are introduced, later stages progressively recover finer temporal dynamics without destroying previously learned structure. This telescoping improvement is characteristic of iterative solvers: new blocks act as corrective refinements rather than re-learning the solution from scratch. The monotonic reduction in rMSE across stages provides empirical evidence that residual depth corresponds to successive refinement steps.

**Residual blocks operate in the small-step regime.** Figure 7 reports the relative transformation magnitude $\|T(z_k)\|/\|z_k\|$ per block for the convection problem. Standard PINNs exhibit consistently larger ratios, indicating aggressive transformations that amplify latent representations and exacerbate anisotropy across depth. In contrast, ResPINNs maintain ratios near unity that decrease with depth, indicating that each block applies a small correction around the identity. This behavior directly supports the Jacobian neutrality analysis in the main text: when updates are small, per-block Jacobians remain close to identity, suppressing spectral growth and stabilizing gradient propagation across depth.

**Alignment patterns and flow mismatch.** Figure 8 shows the cosine alignment between block updates and local loss gradients. Vanilla PINNs oscillate between positive and negative alignment across blocks, indicating inconsistent descent behavior and frequent uphill updates. This oscillatory pattern is a manifestation of *flow mismatch*, where layerwise transformations do not reliably align with descent directions. ResPINNs, by contrast, remain close to neutral alignment throughout depth. Importantly, near-zero cosine alignment does not indicate ineffectiveness; rather, it reflects that residual blocks primarily act as stabilizers that preserve conditioning while enabling multi-step refinement. Combined with the small update magnitudes in Figure 7, this shows that ResPINNs decouple stability from aggressive descent, allowing global error reduction to emerge from many well-conditioned incremental steps.

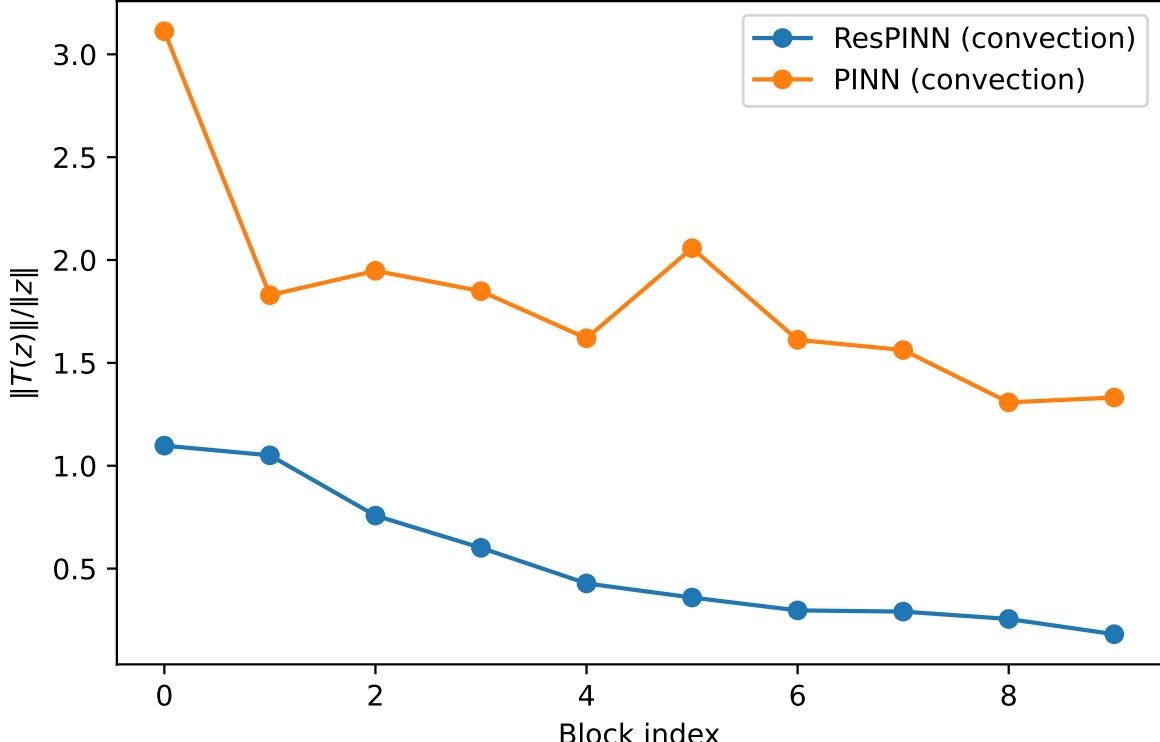

*Figure 7.* Relative transformation magnitude $\|T(z_k)\|/\|z_k\|$ per block for the convection problem. ResPINNs keep ratios near unity, suppressing spectral growth and stabilizing gradient flow. In contrast, PINNs amplify inputs more strongly, reflecting anisotropy and poor conditioning.

## J. More On Error and Solution Maps

This section provides qualitative visualizations that complement the scalar metrics reported in Section 5.2 and Table 2. For PINNs, low residual loss at collocation points can still coexist with substantial *global* error between points, especially in time-dependent problems where condition propagation is fragile. Plotting the predicted solution fields together with pointwise error maps therefore offers a direct probe of the failure modes discussed in the main text: (i) *solution drift* over time and (ii) *structured error bands* that indicate mismatched propagation dynamics rather than purely local inaccuracies.

**Wave: suppressing structured drift.** Figure 10 compares predicted solutions (top row) and pointwise errors (bottom row) on the 1D wave equation. The vanilla PINN exhibits coherent error structure that spans broad regions of the domain, consistent with global drift despite satisfying constraints at training locations. Sequence-based baselines reduce some error but still display non-uniform, anisotropic error regions. In contrast, ResPINN produces a solution visually closest to the reference and an error map with markedly reduced amplitude and less coherent structure. This qualitative pattern supports the Jacobian-neutrality and refinement interpretations: stable, near-identity updates preserve information across depth, preventing the accumulation of systematic drift.

**Reaction: improving condition propagation in stiff nonlinear dynamics.** Figure 11 shows the analogous comparison on the reaction problem, which is known to induce failure modes due to its nonlinear dynamics and sharp interior transition. The vanilla PINN displays pronounced error concentration (especially in regions where the solution changes rapidly), indicating that the learned dynamics do not faithfully propagate the initial profile through time. Residual-flow models substantially suppress these errors, and ResPINN yields the most uniform and lowest-amplitude error map among the compared architectures. Importantly, the improvement is not merely a reduction in average error: the spatial organization of the error changes, with ResPINN eliminating large contiguous error regions that are characteristic of condition-propagation failures. Together, Figures 10 and 11 show that residual flows improve global solution fidelity and reduce structured drift, consistent with the stabilization mechanisms analyzed in the main text.

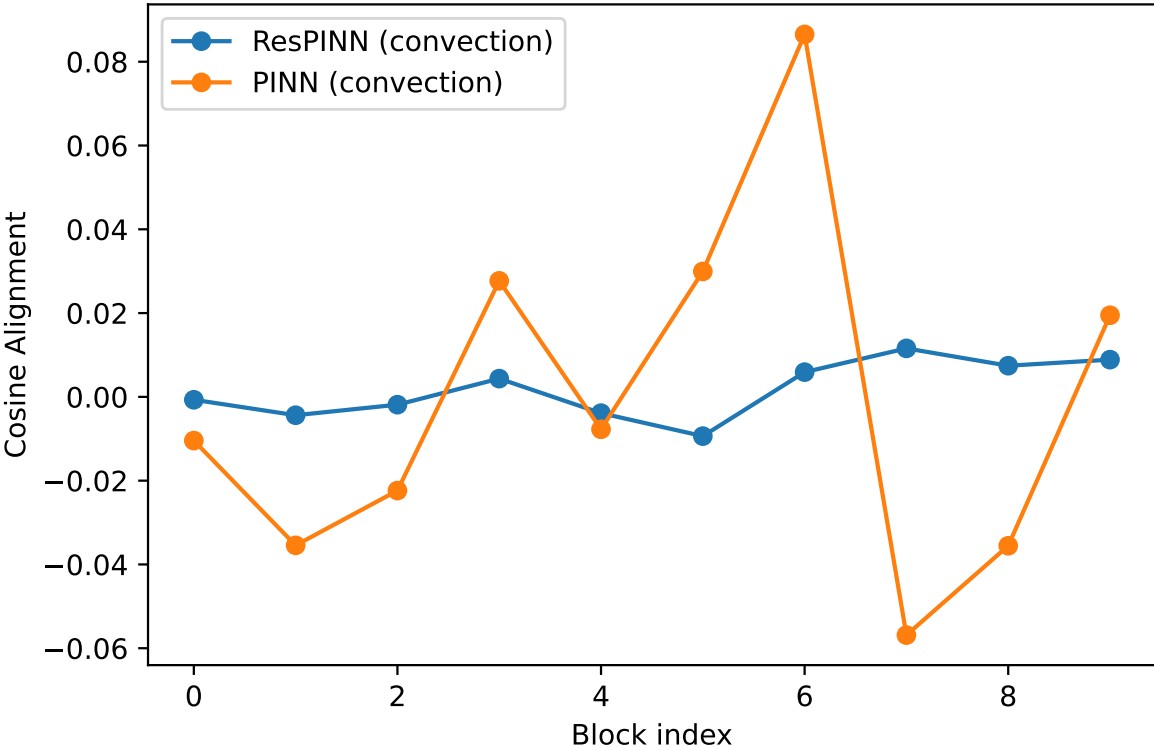

*Figure 8.* Cosine alignment between block updates and local loss gradients for the convection problem. ResPINNs remain close to zero, indicating residual updates act primarily as stabilizers rather than directly following descent directions. PINNs oscillate between positive and negative values, reflecting inconsistent alignment and unstable propagation.

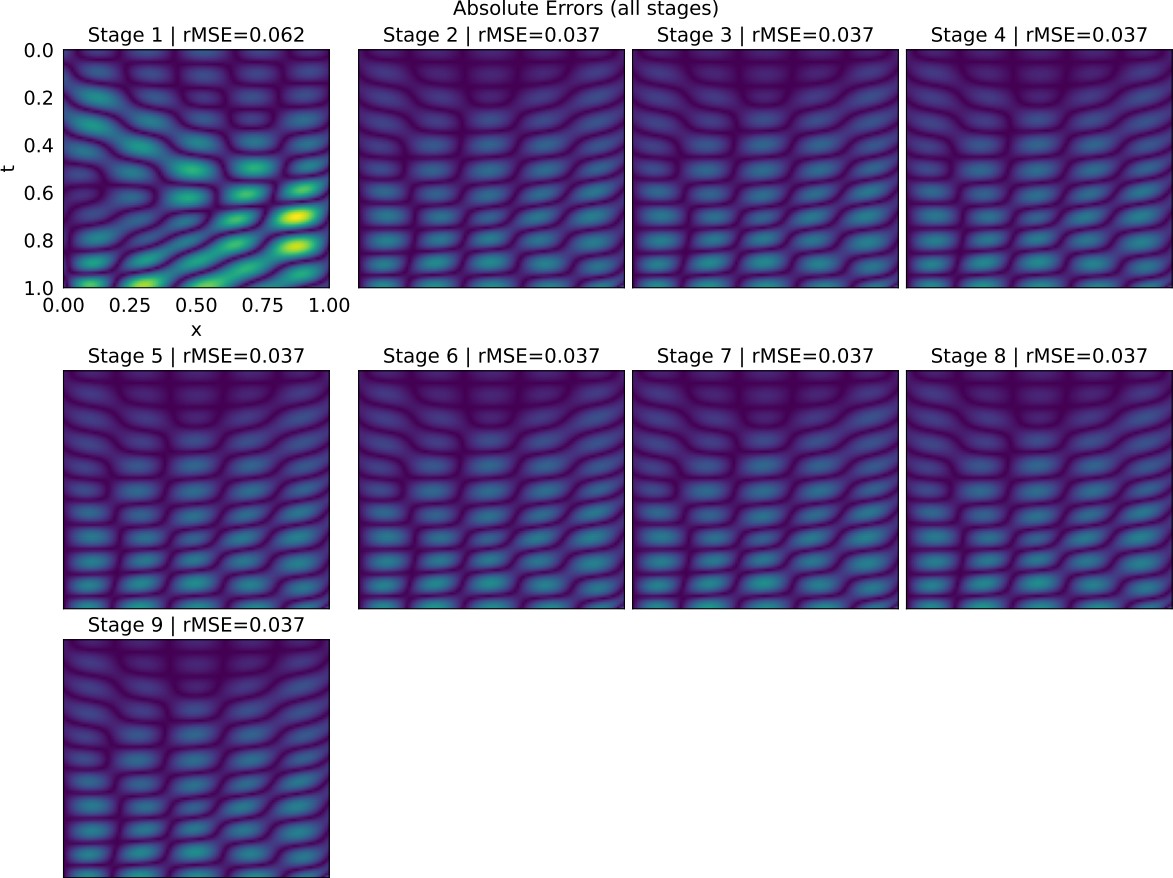

*Figure 9.* Absolute Errors across blocks on wave PDE.

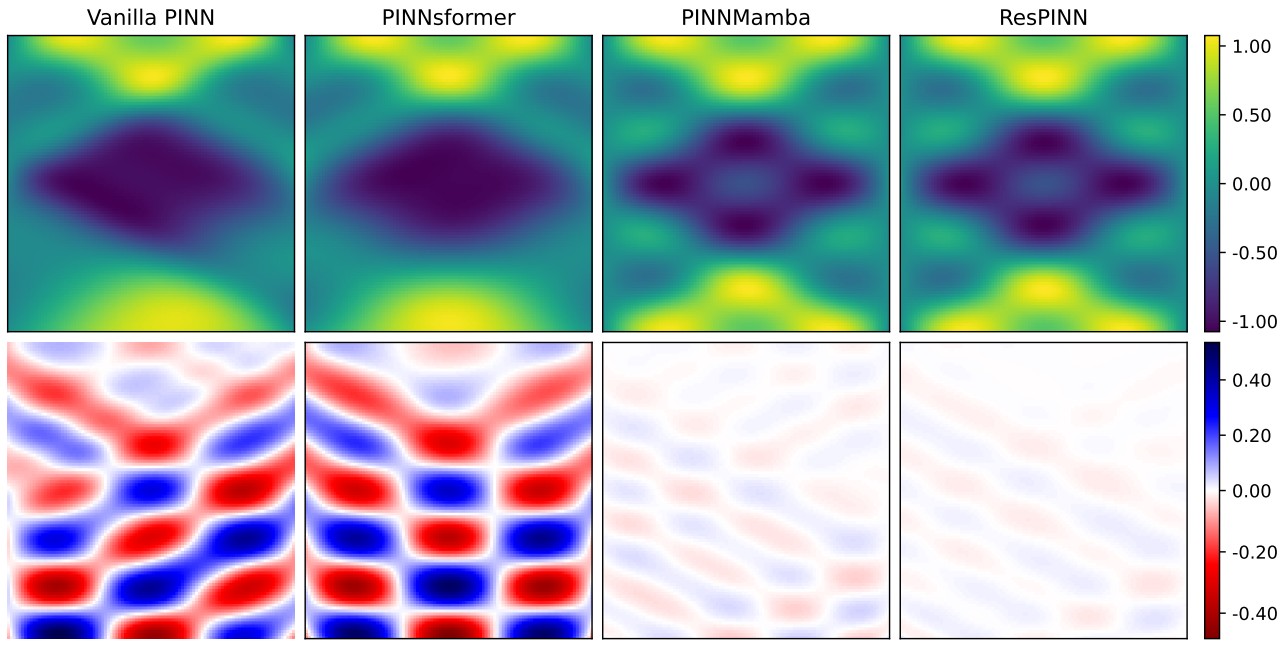

*Figure 10.* Qualitative comparison on 1D wave PDE. Top: predicted solutions. Bottom: pointwise errors.

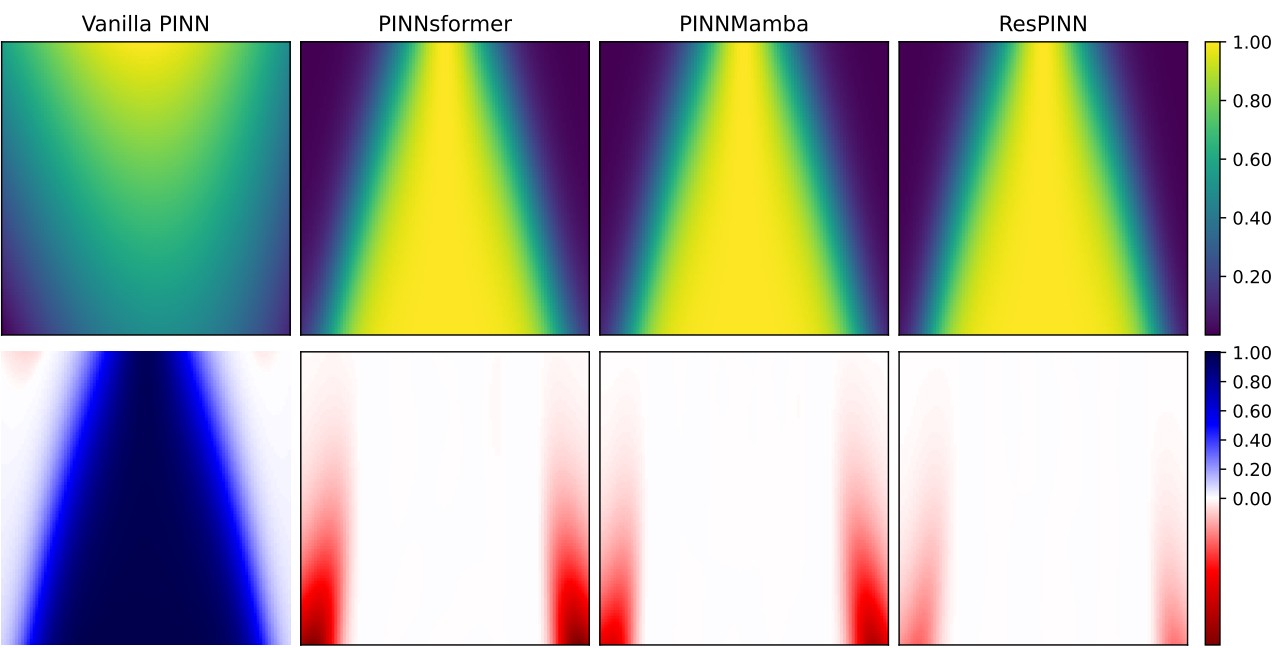

*Figure 11.* Qualitative comparison on 1D Reaction PDE. Top: predicted solutions. Bottom: pointwise errors.

