# OpenReview forum: "Physics-Informed Residual Flows"
_ICML.cc/2026/Conference — ICML 2026 regular_

### Official Review · Reviewer_uPjA · 2026-02-18

**Soundness:** 3
**Presentation:** 2
**Significance:** 2
**Originality:** 2
**Overall Recommendation:** 2
**Confidence:** 5

**Summary:**

This paper proposes "ResPINNs," an architecture designed to mitigate gradient shattering and "flow mismatch" in Physics-Informed Neural Networks (PINNs). The authors reformulate PINN training as a "Residual Flow" in latent space, arguing that iterative refinement via residual connections maintains near-identity Jacobians and aligns updates with descent directions. Empirical evaluations are conducted on standard PDE benchmarks and the PINNacle dataset, comparing the method against MLPs and sequence-based models like PINNsFormer and PINNMamba.

**Compliance With Llm Reviewing Policy:**

Affirmed.

**Key Questions For Authors:**

1. Can you mathematically distinguish "Flow Mismatch" from standard optimization difficulties? Is this a distinct pathological phenomenon unique to the physics-informed loss, or simply a rebranding of the fact that deep networks are hard to train without residual connections?

2. In Section 5, you state that all models, including PINNsFormer and PINNMamba, were trained using L-BFGS. It is well-established that heavy, non-convex architectures like Transformers and State-Space Models typically require gradient-based optimizers (e.g., AdamW) with specific learning rate schedules to converge, and L-BFGS often leads to instability or OOM on such models.

3. Although better than Transformers, ResPINN likely incurs higher training costs and memory usage compared to the standard MLPs it aims to replace, yet this trade-off is not critically analyzed beyond a favorable comparison to the heaviest baselines.

**Limitations:**

Please see the above comments for details.

**Strengths And Weaknesses:**

S1: The paper accurately identifies gradient pathology (shattering) and optimization trajectory divergence (flow mismatch) as critical bottlenecks in training deep PINNs, which is a valid concern in the Scientific ML community.

S2: The ablation study (Table 1) demonstrating that removing attention or SSM modules while retaining residual connections preserves performance is valuable.

S3: The proposed ResPINN demonstrates clear advantages in parameter efficiency and inference speed compared to the heavy Transformer and Mamba-based baselines (as shown in Table 8).

W1: The core contribution is essentially the application of standard Residual Networks (ResNets) to PINNs, rebranded as "Physics-Informed Residual Flows." While the authors attempt to elevate the work through the lens of "iterative solvers" and "flows," the methodology remains a discrete stack of residual blocks. This is distinct from true Neural ODEs or continuous normalizing flows. Applying ResNets to PINNs is an engineering adaptation, not a theoretical breakthrough. Rebranding a well-established technique with sophisticated terminology to claim ICML-level novelty is unconvincing.

W2: The experimental methodology contains a disqualifying flaw. In Section 5, the authors state that all models, including Transformers (PINNsFormer) and State-Space Models (PINNMamba), were trained using the L-BFGS optimizer. It is widely known that heavy, non-convex architectures like Transformers and Mamba require gradient-based optimizers (Adam/AdamW) with specific learning rate schedules to converge. L-BFGS is suitable for small MLPs but catastrophic for these complex models due to memory costs and instability. The extensive "OOM" (Out-Of-Memory) results in Table 5 are a direct consequence of using L-BFGS (which requires storing history for Hessian approximation) on large models. crippling baselines with an inappropriate optimizer to highlight the proposed method's efficiency is scientifically unsound.

W3: The paper compares ResPINN against complex, ill-trained sequence models but ignores simpler, State-of-the-Art (SOTA) improvements to MLPs. For example, MLPs with Fourier Feature Embeddings or Modified MLPs are standard solutions for high-frequency PDEs.

---

> ### Author Rebuttal · Authors · 2026-03-30
>
> We thank the reviewer for the careful reading and detailed feedback. We address each point below.
>
> **W1: Contribution**
>
> We respectfully disagree that this work merely applies ResNets to PINNs. Residual connections are already standard in prior models, including PINNsFormer and PINNMamba. Our contribution is not introducing residuals, but explaining why/how they work, why alternatives fail, and the implications for PINN design.
>
> Concretely, the paper makes three contributions beyond engineering adaptation:
>
> (1) Control diagnostic. Table 1 shows that removing attention from PINNsFormer and SSM modules from PINNMamba, while retaining only residual pathways, preserves or improves performance across all benchmarks. This challenges prior claims that gains arise from sequence modeling (Zhao et al., 2024; Xu et al., 2025a) and instead identifies residual structure as the primary mechanism.
>
> (2) A PINN-specific theoretical analysis. While gradient pathologies are known in deep learning, PINNs differ because their loss differentiates outputs w.r.t. inputs, so Jacobian degradation directly corrupts the training signal. Our flow mismatch analysis (Sec. 3.3, Lemma 3.2, Thm. 3.3) formalizes a PINN-specific failure mode where minimizing residuals at collocation points can still produce global solution drift.
>
> (3) Figures 3 and 7–9 show ResPINN blocks exhibit solver-like behavior, with decreasing update magnitudes and stagewise error reduction. This not only shows that residual connections help, but how they do so, linking to predictor–corrector schemes in numerical analysis. While the reviewer says the method is "distinct from true Neural ODEs", Section 5 explicitly includes O-PINN (continuous-depth with RK4) alongside discrete ResPINN.
>
> Finally, there is strong precedent for work focused on understanding rather than architectural novelty at ICML. For example, Balduzzi et al. (ICML 2017) studied why ResNets work in deep learning; our paper asks the analogous question for PINNs, yielding insights specific to PDE structure.
>
> **W2: Optimizer choice and OOM results in Table 5**
>
> Our paper asks an architectural question: determine whether residual connections, rather than sequence modeling, drive recent PINN gains. This requires an optimizer that sufficiently minimizes the PINN loss so that final accuracy reflects architecture, not optimization. Rathore et al. ICML 2024 show the PINN loss is inherently ill-conditioned and that Adam converges slowly or stalls on the same failure-mode benchmarks we study (e.g., wave PDE, their Fig. 1). Under Adam, all methods perform poorly, obscuring architectural differences. Thus, L-BFGS is not merely conventional but necessary for isolating our question, and it is the protocol used by all baselines we compare against(Zhao et al., ICLR 2024), PINNMamba (Xu et al., ICML 2025), and RoPINN (Wu et al., NeurIPS 2024).
>
> The reviewer correctly identifies that the OOM entries in Table 5 stem from full-batch L-BFGS on higher-dimensional problems. This is a known issue for  PINNs under second-order optimization: PINNMamba report OOM and defer multi-GPU training, while ML-PINN(Gao et al 2025) reduces memory overhead (up to 36% vs. PINNMamba/PINNsFormer). We will add Adam results in the camera-ready version where feasible, but a full study of optimizer schedules and their interaction with residual flows in high dimensions is beyond scope.
>
> **W3: MLP-based PINNs with Fourier embeddings and richer representations**
> We do not ignore these approaches and agree they are important baselines for improving PINNs. Several of our baselines already incorporate such modifications, including enhanced representations (e.g., FLS (Wong et al., 2022) with Fourier features and KANs (Liu et al., 2025)).
>
> **Q1: Formalization of Flow Mismatch**
> We agree that gradient misalignment is a general optimization issue, but flow mismatch is more specific to PINNs due to their multi-objective, soft-constrained formulation(Sec 3.1). Specifically,  the loss $L=\sum_m \lambda_m L_m$ (Lm is PDE, boundary, initial terms or auxiliary terms). Each component induces its own gradient that can conflict or dominate each other. Using gradient alignment in Sec. 3.3, define
> $GA^{(m)}=\langle -\nabla L,\,-\nabla L_m\rangle/\|\nabla L\|\,\|\nabla L_m\|$
> We define *flow mismatch* as the regime where $\exists m:\ GA^{(m)} \le 0$,
> i.e., the aggregate descent direction is not aligned with all constraint gradients. In this case, the loss can decrease via dominance of some objectives while others (including necessary PDE constraints) are not enforced, leading to trajectories that deviate from the solution-consistent path.
>
> **Q2 -> See W2**
>
> **Q3 (Cost vs MLPs).**
> We analyze this trade-off in Table 8 and Appendix H. ResPINNs match MLP memory usage and incur only a modest 1.26x training overhead while significantly improving accuracy. In contrast, sequence-based models such as PINNsFormer and PINNMamba are substantially moreexpensive, with costs up to 7x.

---

> > ### Author Rebuttal · Reviewer_uPjA · 2026-04-05
> >
> > I appreciate the authors' detailed response to my initial concerns. However, before finalizing my assessment, I would like to engage in a deeper technical dialogue regarding several points where the rebuttal appears to conflict with standard practices or the data presented in the manuscript.
> >
> > Can the authors provide evidence that ResPINN maintains its claimed superiority when all models are trained using a more memory-efficient first-order optimizer, which is the standard for the baselines you cite?
> >
> > Beyond rebranding established Deep ResNet principles as "Residual Flows," what specific, non-obvious insight does ResPINN provide that a standard ResNet with Wavelet activations would not already achieve?
> >
> > If the architecture is truly solving the structural failure modes of PINNs, why does it still exhibit such a high sensitivity to arithmetic precision in hyperbolic problems?

---

> > > ### Author Response · Authors · 2026-04-06
> > >
> > > Thank you for engaging with our response.
> > >
> > > ## Q1: Evidence with first-order optimizers
> > > We would like to again highlight that all baselines we cite and compare against *use* second-order optimization (L-BFGS) in their original papers, which is standard in PINNs due to known issues with first-order training. Our setup follows these works for a fair comparison.
> > >
> > > That said, following the reviewer’s request, we trained ResPINN, PINNsFormer, and PINNMamba with Adam (lr 0.001, 5k warmup, 50k steps, exponential learning rate scheduler, same objective/sampling). We report the rRMSE below:
> > >
> > > | Model       | Wave  | Reaction | Convection | NS2dC  |
> > > |-------------|-------|----------|------------|-------|
> > > | PINNsFormer | 0.489 | 0.866    | 0.943      | 1.723 |
> > > | PINNMamba   | 0.571 | 0.094    | 1.112      | 5.504 |
> > > | ResPINN     | 0.068 | 0.053    | 0.031      | 0.036 |
> > >
> > >
> > > As expected, all methods exhibit degraded performance under Adam, consistent with prior work showing that *first-order optimization is insufficient for the ill-conditioned PINN objective* (PINNs have no data supervision, the loss only uses PDE derivatives and the PDE constraints, this makes the loss ill-conditioned). Importantly, ResPINN remains stable and competitive across all settings, while sequence-based models degrade substantially.
> > >
> > > For a comparison. Here is the performance of ResPINN under Adam and LBFGS
> > > | Optimizer | Wave | Reaction | Convection |
> > > |----------|------|----------|------------|
> > > | L-BFGS   | 0.015 | 0.007   | 0.005      |
> > > | Adam     | 0.068 | 0.053   | 0.031      |
> > >
> > > ---
> > > ## Q2:
> > > The term “residual flows” unifies both residual networks and neural ODEs under a common formulation. The key insight is not only that residual connections improve optimization, but that in PINNs they induce a solver-like structure. While residual connections stabilize gradient updates in standard networks, in PINNs they also stabilize PDE derivatives, which form the training signal(An effect we have shown empirically). This leads to an iterative refinement view: solving the PDE through small corrective updates is more effective than learning the solution in a single pass. Empirically, we show that later layers act as PDE error correctors, consistent with this behavior. We also refer to Section 2, where we discuss that PINNs benefit from simple, residual or iterative architectures.
> > >
> > >
> > > ## Q3: Arithmetic Precision
> > > The arithmetic precision sensitivity in hyperbolic problems arises from the PDE itself: the PDE involves multiple higher-order derivatives (e.g., wave equations involves 4 higher-order derivatives), and prior work (Xu et al.(2025 NeurIPS)) has shown that PINNs benefit from higher precision in this setting. We would therefore not characterize ResPINNs as unusually sensitive. Under the same FP32 setting, they outperform all baselines, even exceeding baselines  under FP64.

---

### Official Review · Reviewer_sRDx · 2026-03-10

**Soundness:** 3
**Presentation:** 4
**Significance:** 4
**Originality:** 3
**Overall Recommendation:** 5
**Confidence:** 5

**Summary:**

This paper revisits failure modes in Physics-Informed Neural Networks and identifies two structural pathologies: gradient shattering and flow mismatch. To address these, the authors propose ResPINNs, which reformulate PINNs as residual flows that iteratively refine predictions through small corrective steps around the identity. The paper is well written, providing enough theoretical depth via mean-field analysis, and showing extensive empirical evaluations across canonical PDEs and the broader PINNacle benchmark. At this stage I find the paper is interesting for the community and it would be nice to have it in the conference, but still there are improvements that could strenghten the contribution. I summarize below strenghts and weakneeses.

**Compliance With Llm Reviewing Policy:**

Affirmed.

**Final Justification:**

I am in favour of accepting the paper. In the rebuttal they fully addressed my concerns with extra experiments.

**Key Questions For Authors:**

- Figure 3, 5: Could you also show the Jacobian spectra and alignment diagnostics against PirateNet, which also uses skip connections? This would help disentangle whether the observed benefits are specific to ResPINN's architecture or a more general property of residual connections.
- Table 8: Could you report overhead numbers for more than just the Wave PDE? Computational costs may differ significantly across problem types, especially for stiffer or higher-dimensional settings.
- Why do you train with L-BFGS from the beginning? In the PINN literature it is well established that using Adam first and then fine-tuning with a second-order optimizer (L-BFGS) often outperforms L-BFGS alone. Results using first-order optimizers would also be valuable, especially for Table 5, where many OOM failures may be partly attributable to the full-batch nature of L-BFGS in high-dimensional settings.
- Have the authors considered whether the residual flow perspective could be extended to operator learning settings (e.g., FNOs or DeepONets), or is the stabilization mechanism specific to the collocation-based PINN training objective?

**Limitations:**

yes

**Strengths And Weaknesses:**

### Strengths

- The paper introduces a novel view on failure modes for PINNs, not only focusing on gradient shattering but also analyzing how individual network updates contribute to optimization. This theoretical obseration could have a larger impact and foster new research on architectures to improve on this.
- The paper presents an extensive experimental evaluation appreciated, covering both canonical PDEs and the large-scale PINNacle benchmark, with qualitative and quantitative comparisons across many baselines. Results are strong, showing that the method can achive SOTA for different PDEs.
- The ablation study (Table 1) is a very nice example to convey the paper motivation. Showing that removing attention or SSM modules from PINNsFormer and PINNMamba while preserving residual pathways maintains or improves performance is a strong and clean result that supports the core thesis.
- The computational overhead analysis (Table 8) is a valuable addition since it shows that ResPINN can achieve state-of-the-art accuracy with only 1.26× time overhead and no memory increase, compared to 6–9× for sequence-based models. Also, the different precision analysis (Table 7) nicely shows that ResPINN can be used with lower precision and obtain comparable performance.
- The mechanistic diagnostics effectively support the iterative refinement interpretation and go beyond aggregate error metrics.

### Weaknesses

- The claim that residual connections, rather than sequence modeling, are the primary driver of improvement relies heavily on the ablation in Table 1, but the theoretical analysis focuses on gradient shattering and flow mismatch in isolation. A tighter connection between the theory and the ablation results would strengthen the argument.
- Some results in Table 5 (PINNacle) show ResPINN performing worse than vanilla PINN, and many entries are OOM for all methods (probably due to the use of second-order optimizer). The paper does not sufficiently discuss these cases or explain when residual flows are not beneficial. Also the paper only uses second order optimizer, which is not doable in practical scenarios with many collocation points. I would add additional results (same experiments) for first-order optimizers.

---

> ### Author Rebuttal · Authors · 2026-03-30
>
> We thank the reviewer for the careful reading and detailed feedback. We address each point below.
>
> **W1: connection of residual connections and gradient shattering**
>
> Sec. 3.3 already establishes this connection by showing that residual connections improve Jacobian conditioning and mitigate gradient shattering, which directly impacts optimization stability. We will revise the discussion around Table 1 to make this link to the theoretical results more explicit.
>
> **W2: Optimizer**:
>
> Our use of L-BFGS is mainly for fair comparison across all baselines which also study failure modes in PINNs under a common optimization setting uisng LBFGS. Table 5 indeed highlights challenging regimes where all methods can struggle, and where optimization or memory becomes the dominant bottleneck. Xu et al.(2025) who also studies failure modes in PINNs under the same settings, has suggested that distributed training solves this OOM.
>
> With that being said, as also requested by reviewer pKZB, we evaluated ResPINNs with both first-order (Adam) and second-order (L-BFGS) optimization. observe that ResPINNs remain stable under Adam, albeit with reduced performance.
> | Optimizer | Wave | Reaction | Convection |
> |----------|------|----------|------------|
> | L-BFGS   | 0.015 | 0.007   | 0.005      |
> | Adam     | 0.068 | 0.053   | 0.031      |
>
> For Table 5, we will add Adam results in the camera-ready version where feasible, but a full study of optimizer schedules and their interaction with residual flows in high dimensions is beyond scope.
>
>
>
> **Questions**
>
> **Q1: adding PirateNet to the plots**
>
> We thank the reviewer for the suggestion. We will add the results of PirateNet to the plot in the revised version.
>
> **Q2: Overhead numbers**
>
> We will extend Table 8 to include overhead results on additional PDEs beyond the Wave to provide a more comprehensive comparison.
>
> **Q3: Training with LBFGS**
> As highlighted in W2, our use of L-BFGS is mainly for fair comparison across all baselines. But this also has a deeper meaning to our work. Our paper asks an architectural question: determine whether residual connections, rather than sequence modeling, drive recent PINN gains. This requires an optimizer that sufficiently minimizes the PINN loss so that final accuracy reflects architecture, not optimization. Rathore et al. ICML 2024 show the PINN loss is inherently ill-conditioned and that Adam converges slowly or stalls on the same failure-mode benchmarks we study (e.g., wave PDE, their Fig. 1). Under Adam, all methods perform poorly, obscuring architectural differences. Thus, L-BFGS is not merely conventional but necessary for isolating our question, and it is the protocol used by all baselines we compare against (PINNsFormer (Zhao et al., ICLR 2024), PINNMamba (Xu et al., ICML 2025), and RoPINN (Wu et al., NeurIPS 2024)).
>
> **Q4: Can this perspective be extended to operator learning?**
>
> Yes, this perspective can be extended to neural operators, particularly **physics-informed** neural operators. The same pathology is expected to arise there, since training still involves derivative-based, physics-informed objectives that can induce gradient conflict and misalignment. We reserve a detailed investigation of this extension for future work.
>
> We hope these clarifications address the reviewer’s concerns and make the contributions and scope of the paper clearer.

---

> > ### Author Rebuttal · Reviewer_sRDx · 2026-04-02
> >
> > Thank you for the response, my concerns have been fully addressed and I will increase my score from 4 to 5

---

### Official Review · Reviewer_pKZB · 2026-03-17

**Soundness:** 3
**Presentation:** 3
**Significance:** 3
**Originality:** 3
**Overall Recommendation:** 5
**Confidence:** 3

**Summary:**

The paper investigates the failure modes of physics-informed neural networks (PINNs) and attributes them to gradient shattering and flow mismatch. While recent works have had partial success in addressing these failures, this study argues that the success of those methods should not necessarily be attributed to the complexity of their architectures, but mainly to the incidental presence of residual connections. The authors thus introduce ResPINNs, where the forward pass is formulated as a discrete residual flow that iteratively refines predictions, conceptually bridging deep learning with classical predictor-corrector numerical solvers. By bounding the residual updates to be small, the model enforces local Jacobian neutrality and stabilizes the optimization trajectory without destroying previously learned structures. Consequently, ResPINNs achieve superior accuracy across complex PDE benchmarks while requiring substantially fewer parameters than contemporary architectures.

**Compliance With Llm Reviewing Policy:**

Affirmed.

**Final Justification:**

The authors have adequately addressed my concerns. Although residual connections are not particularly novel in PINNs, the new perspective, and especially the ablation study involving other baselines with residual connections is valuable.

**Key Questions For Authors:**

1. Can the authors provide empirical evidence (e.g., Jacobian cosine similarity measurements at later epochs) demonstrating that gradient shattering remains mitigated by residual connections compared to regular PINNs late in the training lifecycle?
2. Could the authors clarify exactly which minibatch training techniques and optimizers were used to produce the results in Table 6? Furthermore, how does the performance of ResPINN generally compare when using first-order optimizers like Adam versus second-order full-batch methods like L-BFGS?
3. Table 3 reports mixed results for ResPINN and O-PINN. Can the authors provide further insights into when and why the discrete or continuous architecture might be preferable over the other?

**Limitations:**

yes

**Strengths And Weaknesses:**

## Strengths
1. While the inclusion of residual connections in PINNs is not inherently novel, the conceptual connection bridging them to classical numerical solvers and unifying their discrete and continuous (Neural-ODE) forms is a novel and valuable perspective.
2. ResPINN is highly efficient in parameter count compared to contemporary sequence-based baselines and successfully avoids incurring additional memory overhead, while being effective in addressing the failures of PINNs.

## Weaknesses
1. While I appreciate the theoretical motivation behind the use of residual connections, I do not find the theory thorough as to why regular PINNs fail in the first place. Particularly, Theorem 3.1 focuses on the networks at initialization. However, previous empirical analyses of PINN optimization dynamics have shown that the norms of the weights incrementally increase during training. Consequently, it is unclear if the gradient shattering phenomenon persists or is the primary driver of failure late in the training, once these initialization assumptions break down.
2. In relation to the previous point, many other works mitigate PINNs' failures without introducing residual connections. For instance, Fourier feature embeddings are routinely used to mitigate spectral bias and advanced optimization or adaptive sampling techniques are successfully employed to minimize the ill-posed objective. Thus, I am not entirely convinced that gradient shattering can be definitively identified as the primary "structural" cause of PINN failures.
3. The exact proposed architecture is only described in the Appendix. I believe a more detailed description of ResPINN within the main text would significantly help readers, especially by explicitly linking the practical implementation of the residual scale $\alpha$ to the theoretical guarantees provided in Theorem 3.3.

---

> ### Author Rebuttal · Authors · 2026-03-30
>
> **W1: Gradient shattering during and after initialization**
>
> We agree that Theorem 3.1 focuses on initialization and does not fully capture late-stage training dynamics. However, our empirical results (Figure 4) show that even after training, vanilla PINNs exhibit strongly degenerate input–output Jacobians, with most gradients concentrated near zero, in contrast to the analytical solution. This indicates that degraded gradient signals are not merely an initialization artifact but can persist throughout training. We will clarify this distinction and better connect the theory to our empirical observations.
>
>
> **W2: Other Failure Modes in PINNs**
> We agree that gradient shattering is not the sole failure mode in PINNs. Issues such as spectral bias, ill-conditioning, and sampling also play important roles and are discussed in related work. We treat these as complementary mechanisms and explicitly compare against methods designed to address them, including MLP-based PINNs with richer representations such as Fourier-feature approaches (e.g., FLS(Wong et al., 2022)) and KANs ((Liu et al., 2025)).
>
> Our contribution is orthogonal: we isolate the role of gradient pathologies in PINN optimization and show that residual architectures directly mitigate these optimization pathologies. Empirically, ResPINNs achieve higher accuracy than these methods.
>
>
> **W3: Moving the implementation details from the Appendix to the main text**
> We will move a clearer description of ResPINNs from the appendix into the main text. We will also explicitly link the residual scaling parameter to the conditions in Theorem 3.3.
>
>
> **Q1: Plot for the gradient alignment in later eopochs**
>
> We have performed this analysis and observe that residual connections maintain  gradient alignment compared to vanilla PINNs at later epochs. We will include the corresponding cosine similarity plots, including for ResPINNs, in the camera-ready version.
>
> **Q2: Batching and optimizers**
> Following prior work(Zhao et al.(2024,ICLR), Xu et al.(2025a,ICML), Zhongkai et al.(2024, Neurips) ), we use full-batch L-BFGS for all results in Table 6.  The training and batching details are provided in Appendix E.2, with the datasets details in Table 4.
>
> In response to the reviewer’s question, we additionally evaluate we evaluate ResPINNs against Adam using 50k iterations(batch size 2048, learning rate 0.001 with 5k warmup). The resulting RMSE values are:
> | Optimizer | Wave | Reaction | Convection |
> |----------|------|----------|------------|
> | L-BFGS   | 0.015 | 0.007   | 0.005      |
> | Adam     | 0.068 | 0.053   | 0.031      |
>
> Adam yields higher errors across all cases.  While ResPINNs remain stable under first-order optimization, second-order methods like L-BFGS performs better.
>
> **Q3: Recommendation**
>
> We generally recommend ResPINNs due to their simplicity and ease of implementation. While O-PINNs provide a continuous-time perspective, their performance beyond simple settings can depend on the choice and implementation of ODE solvers, introducing additional complexity and variability. In our work, we primarily use O-PINNs as a tool to assess whether the benefits of residual flows extend beyond the discrete setting.
>
> We hope these clarifications address the reviewer’s concerns and make the contributions and scope of the paper clearer.
>
> ------
>
> Zhao et al.(2024, ICLR): PINNsformer: A transformer-based framework for physics-informed neural networks.
>
> Xu et al.(2025a,ICML): Sub-sequential physics-informed learning with state space model.
>
> Zhongkai et al.(2024, Neurips): Pinnacle: A comprehensive benchmark of physics-informed neural networks for solving pdes

---

> > ### Author Rebuttal · Reviewer_pKZB · 2026-04-04
> >
> > Thank you to the authors for their response and the clarifications. My concerns have been addressed.

---

> > > ### Author Response · Authors · 2026-04-04
> > >
> > > Thank you for confirming that all concerns, have been fully resolved and for the positive assessment of our revisions.
> > >
> > > Since the original objection has now been resolved, we would be grateful to know whether any remaining concerns prevent a higher assessment, so that we can further improve the paper accordingly.

---

### Official Review · Reviewer_XpPZ · 2026-03-17

**Soundness:** 3
**Presentation:** 3
**Significance:** 3
**Originality:** 3
**Overall Recommendation:** 4
**Confidence:** 4

**Summary:**

This paper revisits classically understood failure modes in physics-informed neural networks. The authors identify that a number of recent papers have claimed that switching to sequence architectures (state-space models, transformers) significantly improves accuracy, and question whether the improvements are truly due to the sequential structure or simply due to the use of residual connections within these networks. They perform a theoretical analysis of the role of residual connections for general neural network learning, and then conduct an empirical analysis of various architectures on a suite of PDEs, demonstrating that a simple architecture with residual connections matches or outperforms the more complex sequence architectures at a fraction of the cost.

**Compliance With Llm Reviewing Policy:**

Affirmed.

**Final Justification:**

Overall I am left somewhat unsure on the quality of the contribution here. I feel that the theory is quite strangely and sloppily disconnected from the experiments and it is therefore hard to draw a clear conclusion. Overall, I think the authors look at an interesting problem but the paper itself is quite a border case in my mind for acceptance.

**Key Questions For Authors:**

1. To what extent is "gradient shattering" just the standard vanishing/exploding gradient problem? Can the authors clarify the distinction and explain what is specific to the PINN setting?

2. The authors consider a mean field limit $n \to \infty$. Which scaling regime (mean field vs. NTK) are they in, and why? This depends on the initialization and parameterization, which should be specified.

3. Section 3.3 is framed around PINNs, but the loss function and architecture are completely generic. Can the authors explain what about the analysis is specific to PINNs, rather than general neural network learning?

4. In Table 1, dropping the decoder or attention in PINNsformer for convective PDEs dramatically improves performance. Do the authors have any intuition for why?

**Limitations:**

The theoretical analysis is presented as specific to PINNs but is really about residual connections in general neural networks. The authors should be more upfront about this, and ideally should try to connect the theoretical analysis more convincingly with the empirical results. The concern about sparse collocation points seems somewhat overstated for the low-dimensional settings considered.

Overall, I am also left wondering a bit what the main takeaway from the paper should be. Is there a recommended architecture we should use? Is there a clear understanding of why these residual networks work so well for PINNs outside of the sort of handwavy arguments given in the theory section, which don't seem specific to PINNs?

**Strengths And Weaknesses:**

*Soundness:*

The empirical contribution is convincing: the ablation study clearly demonstrates that residual connections, not the sequential structure, are what drive the performance gains in recent PINN architectures. I found this to be a useful and well-executed investigation. The theoretical analysis, however, is less convincing. The authors consider a mean field limit $n \to \infty$, but whether one gets a mean field or NTK limit depends on the specific initialization and parameterization, and the authors do not clarify which regime they are in. Section 3.3 is framed around PINNs, but there doesn't seem to be anything specific to PINNs here; the loss is just an abstract loss function and the architecture is just a neural ODE. The analysis seems to be about the effect of residual connections on neural networks in general, not something specifically tied to physics-informed learning. I also wonder to what extent "gradient shattering" is just the standard vanishing/exploding gradient problem, which residual connections are well-known to fix. The concern about sparse collocation points also seems somewhat overstated, since in practice you typically just randomly sample the domain and these are 2D or 3D problems.

*Presentation:*

The paper is reasonably well-written, but I'm a little confused by the framing in several places. The interpretation at the end of Section 2 presents what seem like standard arguments for residual connections as if they are novel, then reinterprets them in the PDE context. If the core argument is about the role of residual networks for PDE solvers, it should be framed more honestly as such. Similarly, Eq. (2) interprets the residual architecture as a "latent space flow problem," but this is just the neural ODE perspective, which is well-known. The paper would benefit from being more upfront about what is genuinely new versus what is a repackaging of known ideas.

*Significance:*

The practical takeaway is clear and useful: you don't need complex sequence architectures for PINNs; a simple residual MLP works just as well or better at a fraction of the cost. This is a valuable message for the community. The theoretical contributions are less significant, as they apply to general neural networks rather than PINNs specifically.

*Originality:*

The empirical investigation is the main contribution and is well-done. The theoretical analysis, while competent, is not particularly novel and is not clearly tied to the PINN setting.

---

> ### Author Rebuttal · Authors · 2026-03-30
>
> We thank the reviewer for their detailed and constructive feedback. We start with the main takeaway  and then address the reviewer's questions.
>
> **Our recommended takeaway:** PINNs are inherently difficult to train because the loss depends on higher-order derivatives of the network output and multiple soft-contraints, which introduces additional sensitivity and ill-conditioning into the optimization. This makes stable training rely heavily on architectures that preserve well-behaved gradient flow and consistent updates. There has been a precedent that making PINNs deeper could mitigate these problems. We show (In Section 2 and Table 1) that removing  attention/or SSM modules from recent sequence-based models such as PINNsformer (Zhao et al., ICLR 2024), PINNMamba (Xu et al., ICML 2025) maintain or even improve performance. Our main takeaway is that simple residual-style networks are preferable in this setting, as they promote small, aligned updates and avoid the optimization issues introduced by more expressive components like attention or decoders.
>
>
> **Q1:Connection of Gradient Shattering to vanishing/exploding gradients**
>
> Vanishing/Exploding gradient problem is about the gradient magnitude(collapsing to zero or growing uncontrollably large). The Gradient shattering problem: Even when gradients have reasonable magnitude, they can become uncorrelated and noisy across nearby inputs, making them ineffective as a training signal. This is especially harmful for PINNs because many nearby collocation points encode the same local PDE constraint, so learning depends on their gradients aligning rather than canceling out. However, both phenomenon are linked as they come from unstable gradient propagation.
>
> **Q2: Scaling regime**
>
> We consider the mean-field regime, where parameters are scaled such that feature learning occurs throughout training rather than remaining close to initialization as in the NTK regime. This corresponds to standard parameterizations used in practice for PINNs, where weights are not scaled to enforce kernel-like behavior and representations evolve significantly during optimization. We will clarify the initialization and scaling assumptions in the text to make this distinction explicit.
>
> **Q3: Section 3.3 & PINNs**
>
> The formulation in Section 3.3 is generic, but its implications are specific to PINNs through the loss defined in Section 3.1. In PINNs, this loss involves differential operators and multiple objectives, so each update affects both function values and higher-order derivatives. We will revise the text to explicitly reference Section 3.1 to clarify this connection.
>
> **Q4: Why removing attention/decoder can improve the results**
> We believe the decoder and attention layers add expressive power but make PINNs significantly harder to train. They introduce additional nonlinear transformations that can  degrade higher-order derivative signals. Removing them simplifies the optimization.
>
>
> ---
> We appreciate the reviewer’s feedback and believe the added clarifications directly address each point.
>
> ------
> References:
>
> Zhao et al.(2024,ICLR): PINNsformer: A transformer-based framework for physics-informed neural networks.
>
> Xu et al.(2025,ICML): Sub-sequential physics-informed learning with state space model.

---

> > ### Author Rebuttal · Reviewer_XpPZ · 2026-04-05
> >
> > I thank the reviewer for their reply and their justification. While I think the paper has its merits, I still believe the theory is quite strangely disconnected from the rest of the work and that the paper needs some serious revisions for it to be of the quality necessary for publication in ICML. After reading the other reviews, I have decided to keep my score fixed at a 4.

---

### Decision · Program_Chairs · 2026-04-30

**Decision:**

Accept (regular)

**Comment:**

This paper first analyzes the failure modes of PINNs and identifies two main causes: gradient shattering and flow mismatch. The authors then provide a theoretical analysis of residual connections in PINNs, explaining why residual connections can lead to better optimization behavior in PINNs, and validate these findings on datasets.

Overall, three reviewers gave positive recommendations, while one reviewer was quite convinced that the paper should be rejected, indicating that there is some disagreement among the reviewers. More specifically, the reviewers generally found the authors' claim that introducing residual connections can improve the performance of PINNs to be reasonable. Some reviewers felt that the theoretical analysis and observations provide a new and interesting perspective on PINN optimization and could be insightful to readers. Others considered the experiments to be fairly comprehensive and convincing.

At the same time, the reviewers also raised several concerns. On the theoretical side, the main issues include that the analysis is not sufficiently tightly connected to PINNs and PDE solving, that there is a gap between the theory and the experiments, that the explanation of PINN failure modes is not comprehensive enough, and that the algorithmic improvement is not sufficiently novel compared with prior work such as ResNet. Some of the other concerns were addressed in the rebuttal.

Taking the reviewers' comments together, I believe that the paper's perspective on understanding optimization issues can provide useful insight into the development of PINNs, and I am inclined to recommend acceptance. At the same time, the authors should make improvements to the paper in response to the issues raised during the rebuttal process.